

# Early-life diet does not affect preference for fish in herring gulls (*Larus argentatus*)

Emma Inzani[1], Laura Kelley[1], Robert Thomas[2] and Neeltje J. Boogert[1]

[1] Centre for Ecology and Conservation, University of Exeter, Penryn, Cornwall, United Kingdom
[2] Organisms and Environment Division, Cardiff School of Biosciences, Cardiff University, Cardiff, Wales, United Kingdom

## ABSTRACT

Urban populations of herring gulls (*Larus argentatus*) are increasing and causing human-wildlife conflict by exploiting anthropogenic resources. Gulls that breed in urban areas rely on varying amounts of terrestrial anthropogenic foods (*e.g.*, domestic refuse, agricultural and commercial waste) to feed themselves. However, with the onset of hatching, many parent gulls switch to sourcing more marine than anthropogenic or terrestrial foods to provision their chicks. Although anthropogenic foods may meet chick calorific requirements for growth and development, some such foods (*e.g.*, bread) may have lower levels of protein and other key nutrients compared to marine foods. However, whether this parental switch in chick diet is driven by chicks' preference for marine foods, or whether chicks' food preferences are shaped by the food types provisioned by their parents, remains untested. This study tests whether chick food preferences can be influenced by their provisioned diet by experimentally manipulating the ratio of time for which anthropogenic and marine foods were available (80:20 and *vice versa*) in the rearing diets of two treatment groups of rescued herring gull chicks. Each diet was randomly assigned to each of the 27 captive-reared chicks for the duration of the study. We tested chicks' individual food preferences throughout their development in captivity using food arrays with four food choices (fish, cat food, mussels and brown bread). Regardless of the dietary treatment group, we found that all chicks preferred fish and almost all refused to eat most of the bread offered. Our findings suggest that early-life diet, manipulated by the ratio of time the different foods were available, did not influence gull chicks' food preferences. Instead, chicks developed a strong and persistent preference for marine foods, which appears to match adult gulls' dietary switch to marine foods upon chick hatching and may reinforce the provisioning of marine foods during chick development. However, whether chicks in the wild would refuse provisioned foods, and to a sufficient extent to influence parental provisioning, requires further study. Longitudinal studies of urban animal populations that track wild individuals' food preferences and foraging specialisations throughout life are required to shed light on the development and use of anthropogenic resource exploitation.

Corresponding author
Emma Inzani, eli204@exeter.ac.uk

## INTRODUCTION

Living alongside humans presents many challenges for wildlife, as their natural habitat changes at an unprecedented rate. Animals have to cope with, and adapt to, differences between natural and urban landscapes in the accessibility and abundance of resources essential to survival. Urban areas have degraded or replaced coastal areas, woodland, grassland and estuarine floodplains that used to provide animals' natural breeding habitat and prey. Urban areas also hold the potential risk of predation by unfamiliar predators (mainly domestic cats and other non-native species), and of human-wildlife conflict (*Beckerman, Boots & Gaston, 2007*; *Concepción et al., 2015*; *McKinney, 2006*; *Sol et al., 2014*; *Soulsbury & White, 2015*). Although urban areas have fewer natural resources, they can harbour many anthropogenic resources. These include breeding and roosting spaces on/in buildings mirroring natural spaces (*e.g.*, flat roofs as cliff tops for gulls, attics as caves for bats), as well as shelter from natural predators and harsh environmental conditions (*Sol, Lapiedra & González-Lagos, 2013*; *Soulsbury & White, 2015*). People also feed wildlife and leave food waste on the street, in bins, and in rubbish dumps, which can be easily accessible for animals and can become a reliable food source (*Bateman & Fleming, 2012*; *Coulson, 2015*; *Soulsbury & White, 2015*; *Cox & Gaston, 2016*; *Real et al., 2017*). Being behaviourally flexible, a dietary generalist, and making use of increasingly abundant anthropogenic foods can be beneficial for animals and allow colonisation of, and persistence in more urbanised areas (*Bolnick et al., 2003*; *Araújo, Bolnick & Layman, 2011*; *McCleery, 2015*; *Soulsbury & White, 2015*; *Galbraith et al., 2017*; *Callaghan et al., 2019*).

A potential risk of consuming anthropogenic food is that it may not be nutritionally appropriate for the animal. This appears to be particularly important for specialist marine top predators such as Steller's sea lions (*Eumetopias jubatus*, *Österblom et al., 2008*), northern gannets (*Morus bassanus*, *Votier et al., 2010*), black-legged kittiwakes (*Rissa tridactyla*, *Romano, Piatt & Roby, 2006*) and tufted puffins (*Fratercula cirrhata*, *Romano, Piatt & Roby, 2006*); shifting from consuming scarce, high-quality foods such as pelagic fish to consuming prey that is more abundant but of lower nutritional and/or energetic quality, such as fishery discards of demersal fish, has been associated with negative fitness outcomes. For example, adult Cape gannets (*Morus capensis*) that relied on fishery boat discards were not disadvantaged themselves, but struggled to rear chicks to fledgling more so than those adults that caught their own prey (*Grémillet et al., 2008*). This may be due to fishery discards differing in fat, protein and micronutrient composition as compared to pelagic fish, as found by previous seabird studies (*Spaans, 1971*; *Votier et al., 2010*; *Oro et al., 2013*; *van Donk et al., 2017*; *Sotillo et al., 2019*).

In contrast, generalist foragers, with broader dietary ranges, such as various gull species, might be more likely to benefit from anthropogenic access to energy-dense foods, including food that is terrestrial in origin, such as animal carcasses, as well as plant-based food waste from domestic and agricultural sources. Silver gulls (*Larus noveaehollandiae*) for example showed no ill effects of consuming terrestrial anthropogenic foods on adult body mass or condition (*Auman, Meathrel & Richardson, 2008*), while yellow-legged gulls (*Larus michahellis*) exhibited a reduction in body mass following refuse dump closures

(*Steigerwald et al., 2015*). Foraging on terrestrial anthropogenic refuse or agricultural waste has even been shown to increase population sizes and lead to range expansion of gull species such as herring gulls (*Larus argentatus*), kelp gulls (*Larus dominicanus*), lesser black-backed gulls (*Larus fuscus*) and yellow-legged gulls (*Pons, 1992*; *Oro, Bosch & Ruiz, 1995*; *Duhem et al., 2008*; *Lisnizer, Garcia-Borboroglu & Yorio, 2011*; *Camphuysen, 2013*). However, the effects on chick growth and fledging success appear to be mixed; studies of kelp gull (*Lenzi et al., 2019*), glaucous gull (*Larus hyperboreus*; *Weiser & Powell, 2010*) and yellow-legged gull (*Steigerwald et al., 2015*) chicks found positive effects on fledgling success as a result of increased parental access to terrestrial and anthropogenic food sources, and yellow-legged gulls and herring gulls produced larger eggs (*Steigerwald et al., 2015*; *Serré et al., 2022*). In contrast, other studies have reported that gull pairs where one or both parents provided more anthropogenic and terrestrial foods than marine-sourced food had lower productivity and produced chicks with lower body mass (*Annett & Pierotti, 1999*; *O'Hanlon, McGill & Nager, 2017*; *Sotillo et al., 2019*). Furthermore, herring gulls raised larger broods when they provisioned their chicks with more marine foods (including pelagic and intertidal prey), compared to those that provisioned less marine and more terrestrially-sourced foods (*O'Hanlon, McGill & Nager, 2017*). These mixed results may be due to the fact that the type and nutritional content of terrestrial (*e.g.*, grain or chips) *vs.* marine foods (*e.g.*, fish or mussels) obtained depend on breeding colony location and the associated foraging range of the gulls during breeding (*O'Hanlon, McGill & Nager, 2017*; *van Donk et al., 2017*; *Enners et al., 2018*). Thus, high-quality terrestrial food might, in some cases, generate higher reproductive success than low-quality marine food.

Within generalist gull species, individuals have been found to differ considerably in their foraging habits (which also vary with age, seasons, location and breeding status); some individuals specialise on terrestrial foods like refuse, grains, invertebrates in fields, human refuse or intertidal or marine prey (*Lisnizer, Garcia-Borboroglu & Yorio, 2011*; *Steenweg, Ronconi & Leonard, 2011*; *Davis, Elliott & Williams, 2015*; *Gyimesi et al., 2016*; *O'Hanlon, McGill & Nager, 2017*; *Peterson, Ackerman & Eagles-Smith, 2017*; *Mendes et al., 2018*; *Langley et al., 2023*). In contrast, others are generalists and forage on a wide range of foods in both marine and terrestrial habitats (*Brousseau, Lefebvre & Giroux, 1996*; *Annett & Pierotti, 1999*; *Steenweg, Ronconi & Leonard, 2011*; *Davis, Elliott & Williams, 2015*; *Nager & O'Hanlon, 2016*; *O'Hanlon, McGill & Nager, 2017*; *van Donk et al., 2017*; *Peterson, Ackerman & Eagles-Smith, 2017*; *Mendes et al., 2018*; *Spelt et al., 2019*, *2021*; *Westerberg et al., 2019*; *Langley et al., 2021*; *Pais de Faria et al., 2021*). Such inter- and intra-specific variation in foraging strategies, in terms of the different types of food exploited, can be shaped by a range of factors, from differences in life history traits, sex, individual personality traits like explorativeness and boldness, to variation in one's tendency to rely on individual *vs.* social learning and relative resource profitability and local abundance (*Araújo, Bolnick & Layman, 2011*; *Allen, 2019*; *Bolnick et al., 2003*; *Dall, Houston & McNamara, 2004*).

Juvenile gulls' food preferences and foraging strategies are likely to be influenced by their parents and other conspecifics, given that they receive extended parental food provisioning and live in colonies (*Spaans, 1971*; *Pierotti & Annett, 1991*; *Bukacińska,*

Bukaciński & Spaans, 1996). When captive herring gull chicks were fed a single food type from hatching, they preferred the familiar food over novel food when tested at 6 days old, and some chicks rejected novel foods altogether when these were offered again with no alternative (Rabinowitch, 1968). Variation in individuals' food preferences may thus be influenced by differences in early-life experiences from parental provisioning, as well as individual variation in food neophobia (temporary fear of novel food) and dietary conservatism (a prolonged avoidance) (Marples & Kelly, 1999). At least 75% of the UK's herring gull population is currently breeding in urban areas (Burnell, 2021). Herring gulls often exploit human refuse and food waste (Belant, Ickes & Seamans, 1998; Burnell, 2021; Monaghan, 1979; Raghav & Boogert, 2022; Rock, 2005), as it can be temporally and spatially more predictable and abundant than foraging for marine prey, and can provide more calories for less time and effort spent foraging (Pierotti & Annett, 1991; Belant, Ickes & Seamans, 1998; van Donk et al., 2017). However, urban-nesting in the UK only commenced in the early 1900s (Monaghan & Coulson, 1977), and parent herring gulls are often observed to switch to a more marine, higher-protein diet when their chicks hatch, possibly to better meet the nutritional requirements of chick development and growth (Bukacińska, Bukaciński & Spaans, 1996). Herring gull chicks may thus have a pre-existing preference for marine foods, and integrate terrestrial anthropogenic foods into their diet later, through either personal experience or parental provisioning. How urban herring gull chicks acquire their dietary preferences, is, however, poorly understood.

The aim of this study was to test whether herring gull chicks have individual food preferences and whether those can be influenced by early-life dietary experience. We repeatedly tested whether young captive-reared herring gull chicks preferred marine or terrestrial foods shortly after arrival to captivity, and then while being reared on either a predominantly (i) marine or (ii) a terrestrially-sourced diet. Our dietary manipulation reflects the extremes in the range of foods that herring gulls have been observed to provision their chicks in the wild (Pierotti & Annett, 1991; Serré et al., 2022), to explore whether chicks would develop a preference for those foods that they were provisioned with. As described above, we predicted that herring gull chicks might initially prefer marine foods. However, we hypothesised that chicks in the terrestrial diet group would shift their preference to consume more terrestrial than marine foods due to habituation to their experimental rearing diet. Each rearing diet also contained both high- and low-protein food types. We predicted that chicks in both treatment groups would consume high-protein options first, as this would be the most beneficial choice to maximise growth and development (Spaans, 1971; van Donk et al., 2017; Sotillo et al., 2019). Finally, we tested for consistent intra- and inter-individual differences in food preferences within treatment groups, which could result from a combination of genetic factors, maternal effects and individual experiences prior to our experiment (Monaghan, 2007).

## METHODS

### Ethics

This study was approved by the University of Exeter, College of Life and Environmental Sciences, Penryn Ethics Committee (eCORN002962 9.1). GPS tagging (as part of a

concurrent study and not included here) and marking of chicks with leg rings and non-toxic temporary dye were approved by the British Trust for Ornithology Special Methods Technical Panel on behalf of the Joint Nature Conservation Committee & Natural England (application licence numbers: 11962 and 11963). The total time that each chick was handled during ringing did not exceed 20 min, except for seven chicks that were GPS-tagged (GPS back-mounted thoracic weak-link harness), where handling time did not exceed 50 min. Avian specialist vets monitored the health and welfare of the gull chicks throughout the study and ascertained that chicks were fit for release. Chicks were kept in captivity until they demonstrated strong flying capabilities in the pens and were confirmed fit for release by the vet. Chicks that were included in this study were kept in captivity for the purpose of rescue and rearing to independence, and their inclusion in this experiment did not hinder or delay their release from captivity.

## Chick husbandry during captivity and release

The herring gull chick rehabilitation facilities were at a private residence near Troon, Cornwall, which contained unimproved grassland areas on the property and was surrounded by improved grassland fields within 1 km.

All herring gull chicks brought into captivity for rearing and rehabilitation were first assessed by avian specialist veterinarians to check their condition. The chicks were apparent 'orphans' who could either not be reunited with their parents, placed back into their nest, or their exact origin was unidentifiable, thus they were unlikely to survive without intervention. All chicks were found and brought into the rehabilitation facilities within 24 h of rescue from towns across Cornwall, the majority from residential roofs. Over seventy percent of the county of Cornwall is agricultural land, while Cornwall also has 697 km of coastline. It therefore seems likely that most of the chicks would have been provisioned with a mixture of marine and terrestrial foods while still at the nest. However, other than their rescue location, we do not have any information regarding their early-life or dietary experiences in the wild before they entered the rehabilitation facilities.

Chick ages were estimated by ELI (an experienced ornithologist) on admission to captivity, according to their weight, feather development and other physiological features (*e.g.*, presence or absence of an egg tooth). The mean chick age ± SD was estimated to be 10 days old ± 8 days ($n = 27$). All chicks included in our study were assessed by a vet to be healthy, with no major injuries or other health concerns. Any chicks brought in with treatable injuries or illnesses were not included in the study cohort. Some non-study chicks were present in the enclosures with the study chicks. Any chicks showing illness or distress during rehabilitation were immediately taken to a vet and received medical treatment. Two study chicks developed a respiratory disease whilst in care and were temporally separated and removed from the study until they had fully recovered and were confirmed healthy by a vet, and were returned to the study cohort. Both were from the Marine diet treatment group; one chick missed a 24 h test, and the other the day five test (see below for details of treatments and tests). As their experimental diets and apparent appetites were unchanged during this period, we decided not to remove them from the study sample. Chicks were weighed daily until ca. 30 days of age, then at least once weekly to check on

their health and development. Tarsus length was also measured at least weekly to quantify skeletal growth. Handling times were ca. 5 min/day for small chicks and ca. 2 min/week for chicks >30 days old.

Chicks <5 days old were initially kept in an incubator together and fed with tweezers approximately every 2 h during the day, until they were reliably feeding independently. Chicks 5–25 days old were housed indoors overnight in groups of three in cages (0.6 m length × 0.5 m width × 0.5 m height) that were raised off the floor for safety reasons. From ca. 10 days old, they were housed in indoor ground pens (0.9 m × 0.9 m × 0.5 m height) during daylight hours (ca. 10 h a day), with up to 9 chicks in one pen with *ad lib* food, water and environmental enrichment (*e.g.*, seaweed, cardboard boxes, water trays) until ca. 25 days old. At night, chicks were still housed in their groups of three in the raised overnight cages until ca. 25 days of age. Each chick was marked with a spot of non-toxic sheep marker-spray on downy feathers (approved by the British Trust for Ornithology (BTO) Special Methods Technical Panel (SMTP)) to allow for individual identification whilst <15 days old. Once the chicks were >15 days old, they were fitted with temporary poultry colour rings on their tarsus for ID purposes.

From ca. 25 days of age, when chicks' body feathers had grown sufficiently to make them waterproof, chicks were moved into partially sheltered outdoor aviaries (ca. 2 m × 4.6 m × 2.5 m) for a week to acclimatise and then into outdoor flight pens (ca. 7 m × 3.5 m × 2.5 m) with other chicks assigned to the same dietary treatment (up to 15 chicks per pen). Outdoor enclosures were on grass with netted roofs and included a paddling pool, perches, foraging enrichment (*i.e.*, scattered dried mealworms (*Tenebrio molitor* larvae) and fresh seaweed), shelters, and *ad lib* food and water. Once the chicks were assessed by a vet and deemed fit for release at ca. 50 days old, they were ringed with a permanent BTO metal ring on one leg and an alpha-numeric coded blue plastic colour ring on the other. Prior to release, each chick was weighed, measured for minimal tarsus length (*Caravaggi et al., 2021*), head length (measured from the back of the skull to the tip of the upper mandible), bill length (from the tip of upper mandible to where the base of the upper mandible meets feathers) and bill depth (measured at the gonys angle on the bill) and scored for aggression (number of bites/pecks received by handler during ringing).

The first, older group of 15 chicks were soft-released on 26[th] July at the average age (mean ± SD) of 69 ± 12 days old, by removing the netted tops of their enclosures to allow the gulls to leave by flight during the day. Video recordings showed that all the juveniles flew out of the pens within several hours of the netting being removed, and four individuals left within the first half an hour. However, the chicks remained at the rearing site and spent the night as a group in the surrounding field. Three of these chicks were predated by red foxes (*Vulpes vulpes*). A further two of the released chicks were found around the pens trying to enter them again so they were returned to captivity to be re-released at a later date, while a third was also seen but chose to fly away. The remaining nine gull chicks were not found onsite and are presumed to have been released successfully. The second group of 14 chicks (including the two returned to captivity) were released later at a local reservoir and nature reserve (Stithians Reservoir, Cornwall *SW 7325 3691*) on August 23[rd] and 24[th], when their mean age (±SD) was 71 ± 12 days. One study chick was retained as it tended to
approach humans and was released on 19[th] September once it showed more appropriate wariness towards people.

## Experimental dietary treatments

Chicks ($n = 27$) were assigned to one of two experimental dietary treatment groups; thirteen to the 'Terrestrial' group and fourteen to the 'Marine' group, where the dietary treatments resembled opposite extremes of gull parental provisioning strategies observed in the wild (*van Donk et al., 2017*; *Sotillo et al., 2019*; *Serré et al., 2022*). Chicks were provided with foods categorised as terrestrial or marine in an 80:20 ratio. All chicks were exposed to all foods offered, to avoid food neophobia (*Rabinowitch, 1968*). Each treatment diet also included high protein and low protein food, with the low protein food accounting for 20% of the wet mass of food provided (see Supplemental Materials Table S1 for the nutritional value of foods). The diet ratios were achieved by presenting the different food types (terrestrial and marine) for set amounts of time throughout daylight hours (ca. 12 h a day), with the order in which food types were presented changing daily, to achieve 80:20 ratios of the food types offered. All chicks were fed *ad lib*; the amount of food provided was proportional to the number of chicks present and their ages, food was always available during the day, and food was added to or exchanged four times a day for small chicks up to 15 days old, and three times a day for older chicks. The number of visits to older chicks was reduced to limit their exposure to people to minimize tameness, which could impact their survival upon release. All non-consumed foods were removed when a food type switch occurred. When all chicks reached ca. 25 days of age and were moved outside, they were provided with terrestrial and marine foods in a 50:50 ratio simultaneously. The latter change in the ratios of food offered was not initially planned, but we observed that chicks were reluctant to eat when not offered marine foods. The chicks prioritised consuming the marine foods when offered, which could have caused some less- competitive chicks to have a reduced food intake, because they refused to eat terrestrial foods.

The high-protein terrestrial food provided was tinned cat food (chicken, turkey, lamb and beef), all flavours in jelly and gravy (Tesco and Asda own-brand cat foods). Chicks <10 days old were provided with cat food chunks of ca. 0.5 cm × 1 cm × 1 cm, while older, larger chicks were given 2–4 cm³ chunks. The lower-protein terrestrial food was diced brown bread, sized ca. 1 cm³ when chicks were small, and 1 cm × 2 cm × 2 cm for larger chicks. We chose bread as the low-protein terrestrial food type as it is commonly consumed by gulls in urban areas from domestic garbage and from people feeding ducks at parks and lakes (*Spaans, 1971*; *Pierotti & Annett, 1991*; *Scott, Duncan & Green, 2015*; *van Donk et al., 2017*). We included bread in the terrestrial dietary treatment group to test whether early-life provisioning with bread would generate a preference for this food. In the marine treatment diet, the high-protein food was whole fish (European sprats *Sprattus sprattus* and Atlantic mackerel *Scomber scombrus*), cut into 1–2 cm³ pieces when chicks were small, while larger chicks were presented with whole sprats that were ca. 7 cm × 2 cm, and pieces of mackerel cut to similar sizes. The lower-protein marine food was pre-cooked mussels (*Mytilus edulis*, Tesco frozen cooked shell-less mussels, ca. 1 cm × 1.5 cm × 2 cm). To compensate for any potential nutritional differences between diets, we added thiamine
(vitamin B1) and vitamin E supplements (Holland and Barrett 100 IU vitamin E capsules, oil removed from capsules, and Holland and Barrett 100 mg thiamine tables, quartered and crushed) at appropriate dosages (vitamin E: 100 IU kg$^{-1}$, thiamine: 25 mg kg$^{-1}$) per kg of fish, and a multi-vitamin and calcium powder supplement (Nutrobal® Vetark) to bread, to ensure normal chick development. These dosages were prescribed by the veterinarian. It was not possible to overdose an individual chick as no one chick could consume all the fish presented in a feed, nor be able to monopolise the food, as it was presented in multiple bowls spaced through their enclosures. The supplements were also only added to the first feed of the day to avoid the potential for overdosing on vitamin E. However, it should be noted that other seabird species can tolerate large dosages exceeding the daily recommended allowance with no apparent side effects (*Crissey, McGill & Simeone, 1998*; *Crissey, Slifka & McGill, 2002*). Thiamine is water soluble and cannot cause overdoses as excess is excreted by the chicks naturally. The Nutrobal® powder was added to the bread in sufficiently small doses that overdosing was not possible.

## Experimental protocols

Immediately upon arrival to captivity, each chick was alternately and randomly assigned to either the Marine ($n = 14$) or the Terrestrial ($n = 13$) diet treatment group. For the experimental tests of chicks' behaviour, chicks were individually placed and tested in ground pens adjacent to their overnight cages (0.9 m length × 0.9 m width × 0.4 m height) in which they were kept during the day, but in the absence of any group mates, objects or shelters. Chicks <15 days old were tested in either the ground pens or a clean overnight cage (0.6 m × 0.5 m × 0.5 m) if they were not yet familiar with the ground pen when < 5 days old. Once outside, chicks >25 days old were tested in an empty flight pen of the same dimensions as, and adjacent to, the flight pens they occupied (ca. 7 m × 3.5 m × 2.5 m). All tests were filmed using a Panasonic HC-V770 video recorder placed outside the enclosures. Chicks were tested at 24–72 h, 5, 10, 15 and 35 days in captivity. All experimental tests were conducted between 7–9:30 am, prior to weighing and *ad lib* feeding. The four food types (terrestrial high-and low-protein food, and marine high- and low-protein food) were offered in experimental food arrays comprising of four grey plastic cat food bowls (CatCentre® 0.35 l bowls). Each bowl contained only 10 g of food to avoid chick satiation. The food bowls were the same as those used for the daily feeding of the chicks and were thus familiar to the test subjects. Some individuals missed some experimental tests for health reasons, but all chicks were tested at least four times.

A baseline experimental test for food preference, which was conducted 24–72 h after arrival into captivity, commenced once the chick did not show any apparent signs of distress, and tested for their baseline preference for marine or terrestrial high-protein foods (*i.e.*, fish *vs*. cat food). After an initial 10 min to habituate to their test enclosure, two food bowls, each containing 10 g of one or the other food type, were presented equidistantly, at approximately 20 cm in front of the chick, for a maximum of 10 min. We measured the latency of the chick to approach and consume (*i.e.*, food visibly eaten or pecked at three times) their first food choice and what that food category was. We also scored whether the chick approached the food/experimenter as the food bowls were being placed inside the

cage or pen (yes or no) as a measure of tameness. Both foods were weighed before and after the test to calculate the amount consumed. The trial was stopped once all food had been consumed, or when 10 min had passed. The presentation side of the two foods (left or right hand side) was randomised. Chicks <3 days old that struggled to feed independently were not tested in this baseline trial.

A protocol similar to the baseline preference test was used on days 5, 10, 15 and 35 to test for potential changes in chicks' food preferences due to the dietary treatments. Chicks were given 10 min to habituate to the test area, after which they were presented with a 2 × 2 grid of food bowls with 10 g of one of the four foods (chopped fish, mussels, cat food and diced bread) in each, and left for 10 min (see Supplemental Material for video of a food preference test trial). The position of each food in the array was randomised in every trial to control for side biases and spatial learning. As before, the food offered was weighed before and after the trial to calculate the amount consumed. For each trial, we recorded the food that the chick consumed first (*i.e.*, food visibly eaten or pecked at three times), the latency to consume the food (time between presentation of the food and it being consumed/pecked at for third time), the second food consumed, and if the chick approached the food as the food bowls were placed into the enclosure (yes or no). All 27 chicks were presented with up to four replicate trials of the food preference tests throughout their time in captivity, but not all chicks chose to participate and consume food during the trials (day 5: $n = 25$; day 10: $n = 26$; day 15: $n = 24$; day 35 (when outdoors and tested in adjacent outdoor test pen): $n = 13$).

All behaviour was scored from video recordings. Videos were scored by ELI and an independent observer blind to the hypothesis being tested to assess the inter-observer reliability of the behavioural scores. To test whether two independent observers agreed on the chicks' food preferences, we calculated Intra Class Coefficients (ICCs). Both the latency for chicks to approach and consume food, and the first food eaten had very high repeatability between scorers (Latency: ICC = 0.917 (95% Cl [0.884–0.941]), $P < 0.001$; First food: ICC = 0.921 (95% Cl [0.887–0.945]), $P < 0.001$).

### Statistical analyses

All analyses were conducted using R version 4.0.3 (*R Core Team, 2020*) and the packages lme4 (*Bates et al., 2015*) and MASS (*Venables & Ripley, 2002*). The error family and link function combination for each generalised linear mixed model (GLMM) conducted was tested to select for the lowest AIC value and smallest residuals.

To determine whether food location within the 2 × 2 arrays affected choice, we performed chi-squared contingency tests. We found that chicks were most likely to choose a bowl from the front of the array (*i.e.*, nearest to them) first ($\chi^2 = 49.50$, df = 1, $n = 88$ trials, $P < 0.001$). However, chicks did not appear to have side biases (left *vs.* right side comparison; $\chi^2 = 0.44$, df = 1, $n = 111$ trials, $P = 0.51$). Neither bowl position nor side influenced chicks' second food choices (bowl at front or back of array: $\chi^2 = 2.65$, df = 1, $n = 74$ trials, $P = 0.10$; left *vs.* right side: $\chi^2 = 0.22$, df = 1, $n = 74$, $P = 0.64$). As food type position was randomised for each trial, bowl position and side were therefore not included in further analyses.

Individual chick weights and tarsus measurements taken over the duration of captivity were compared between the two diet treatment groups using Wilcoxon's tests. Chick weights were compared when chicks were ca. 5 days old ($n$ = 4 chicks per treatment group), 15 days ($n$ = 8 from the marine group, 7 from the terrestrial) and when last weighed recorded prior to release ($n$ = all 27 study chicks). For the final weighing, chick age ranged from 52–79 days due to differences in chick age when received into captivity. Chick tarsus measurements were compared when chicks were ca. 14–16 days old ($n$ = 7 chicks from each treatment group). Tarsus measurements were not taken at younger ages due to time constraints. Chicks' last tarsus measurement prior to release was taken when chick age ranged from 31–58 days old, when all chicks ($n$ = 27) should have reached their growth asymptote (Supplemental Materials, Figs. S1 and S2 for chick weights and tarsus measurements, respectively).

To test whether chicks' first, non-baseline, food choice for marine (fish or mussels) or terrestrial (cat food or bread) food types was influenced by their diet treatment group (Marine vs. Terrestrial), we used a GLMM of the binomial family (link = probit) with diet treatment group as a fixed effect. Included was test day (day 5, 10, 15 and 35) as a fixed effect, to test for the potential influence of increasing chick age and experience on food preferences. Chick ID was included as a random effect to account for repeated sampling. Test trials where no food was consumed were not included in the analysis.

We used Chi-squared tests to determine whether there was a significant preference for, or avoidance of, one of the four foods offered in the chicks' first food choices for each test trial day (day 5, 10, 15 and 35) separately, to avoid pseudoreplication. To test for individual consistency in first and second food preferences, we performed two Intra-Class Coefficient tests (ICCs).

To test whether chicks were preferentially choosing higher protein food (fish or cat food) over lower protein options available (bread or mussels), we used exact binomial tests for each test trial day, excluding the baseline trial, to determine whether chicks chose a higher protein food first (yes or no) more often than would be expected by chance (0.5). We also used exact binomial tests for each test trial day to determine whether chicks were significantly more likely than chance to choose high protein foods consecutively in both their first and second choice (probability of occurring by chance was 0.83).

## RESULTS

### Does dietary treatment affect chick growth?

Chicks in the terrestrial group were significantly lighter at their last weight measurement before release as compared to the marine group (median difference = 110 g, 95% CL [10–215] g; Wilcoxon test, W = 138, $n$ = 14 Marine diet chicks, 13 Terrestrial, $P$ = 0.02). This weight difference measured when 52–79 days old was not apparent when chicks were ≤15 days old (Wilcoxon tests at 5 and 15 days old: $P$ value > 0.05). Similarly, when chicks were 31–58 days old, the tarsi of those in the terrestrial group were significantly smaller than those in the marine group (median difference = 3.5 mm, 95% CL [1.7–4.9] mm; Wilcoxon test, W = 152, $P$ = 0.004). This difference between the diet groups was not quite

**Table 1 *Post-hoc* pairwise comparisons using chi-squared tests for given probabilities (0.25) of each of the four foods consumed first across the food preference trials.**

| Food | Frequency of being consumed first | Expected frequency | Chi-square value | *P* value |
|---|---|---|---|---|
| **Day 5 trial** | | | | |
| Bread | **0** | **6.25** | **8.33** | **0.016** |
| Cat food | 3 | 6.25 | 2.25 | 0.53 |
| Fish | **17** | **6.25** | **24.65** | **<0.001** |
| Mussels | 5 | 6.25 | 0.33 | 1 |
| **Day 10 trial** | | | | |
| Bread | 1 | 6.5 | 6.21 | 0.051 |
| Cat food | 2 | 6.5 | 1.28 | 1 |
| Fish | **13** | **6.5** | **8.67** | **0.013** |
| Mussels | 8 | 6.5 | 0.46 | 1 |
| **Day 15 trial** | | | | |
| Bread | 2 | 6 | 3.56 | 0.24 |
| Cat food | 1 | 6 | 5.56 | 0.074 |
| Fish | **12** | **6** | **8** | **0.019** |
| Mussels | 9 | 6 | 2 | 0.63 |
| **Day 35 trial** | | | | |
| Bread | **0** | **3.25** | **4.33** | **0.015** |
| Cat food | 2 | 3.25 | 0.64 | 1 |
| Fish | **8** | **3.25** | **9.25** | **0.009** |
| Mussels | 3 | 3.25 | 0.03 | 1 |

**Note:**
Significant terms (*P* value < α = 0.05) are highlighted in bold.

significant when chicks were 14–16 days old (Wilcoxon test, W = 40, *n* = 8 Marine chicks, 7 Terrestrial chicks, *P* = 0.06).

## Do herring gull chicks have food preferences, and does the rearing diet influence individual food preferences?

Twenty-three of the twenty-seven chicks participated in the baseline food preference test. Chicks showed no preference for either food type (exact binomial test: number of chicks choosing fish = 13/23; *P* = 0.68). In the later test trials with all four treatment diet foods presented in a 2 × 2 grid, chicks' first food choices did differ from chance (Day 5 trial: $\chi^2$ = 26.68, df = 3, *n* = 25, *P* < 0.001; Day 10 trial: $\chi^2$ = 12.46, df = 3, *n* = 26, *P* < 0.001; Day 15 trial: $\chi^2$ = 14.33, df = 3, *n* = 24, *P* < 0.001; Day 35 trial: $\chi^2$ = 10.69, df = 3, *n* = 13, *P* = 0.01). As shown in Table 1 and Fig. 1, chicks preferred to consume fish first and appeared to avoid bread. Although mussels and fish seemed to be preferred over cat food, and both cat food and mussels appeared to be preferred over bread, these patterns were not significant. We found no influence of the diet treatment group nor the test day on chicks' preferences for marine or terrestrial food types (Table 2; GLMM: *P* > 0.05). Chick ID explained no variation when included as a random effect in this GLMM model, indicating that the variation between individuals in initial food preferences was undetectable
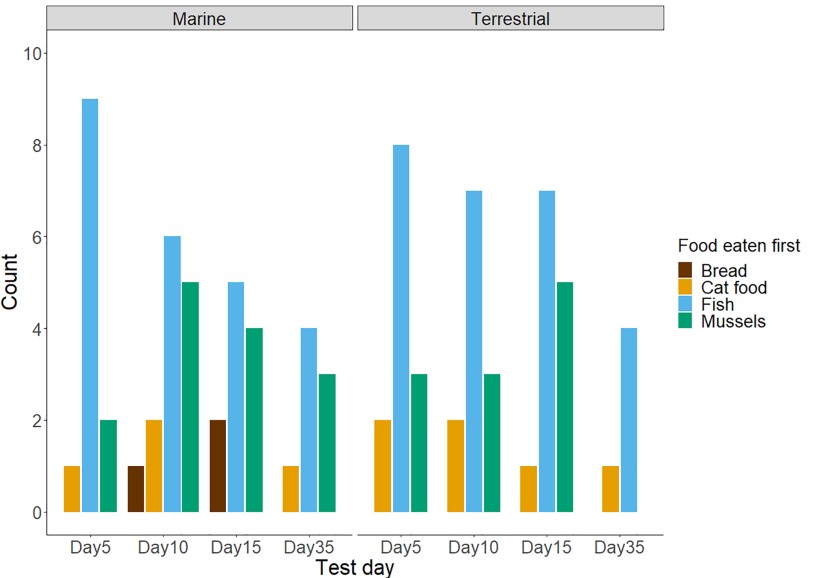

**Figure 1** **The frequency of chicks' first food choices in four test trials (*n* = 88 observations of *n* = 27 chicks).** Not all chicks participated in every test trial: day 5 = 25, day 10 = 26, day 15 = 24, and day 35 = 13 chicks. The bar colour represents the food type. Results for chicks in the Marine group are presented on the left, and results for the Terrestrial group are presented on the right.

**Table 2  Results of a binomial GLMM (link = probit) testing whether herring gull chicks prefer marine or terrestrial foods.**

| | Estimate | Standard error | Degrees of freedom | Z value | *P*-value |
|---|---|---|---|---|---|
| Intercept | −1.15 | 0.37 | 1 | | |
| Diet treatment group | −0.05 | 0.33 | 1 | −0.15 | 0.88 |
| Test day 10 (proxy for chick age and compared to day 5 in intercept) | −0.002 | 0.011 | 1 | 0.70 | 0.49 |
| Test day 15 (proxy for chick age and compared to day 5 in intercept) | 0.02 | 0.46 | 1 | 0.05 | 0.96 |
| Test day 35 (proxy for chick age and compared to day 5 in intercept) | 0.15 | 0.53 | 1 | 0.28 | 0.78 |

**Note:**
Random term: Chick ID variance within the model accounted for = 0, standard deviation = 0. This term had to be included to account for repeated measures of individuals over time.

(Table 2). This might be due to most chicks consistently choosing marine and higher protein foods first in the great majority of trials (Fig. 2A). For chicks' second food choices, fish and mussels were preferred over cat food, while bread remained a rare choice (Fig. 2B).

Chicks' first and second food choices appeared to broadly reflect the amount of each food type that they consumed (Fig. 3). Bread consumption was very low, and only occurred after chicks had consumed all the food in the other bowls. Fish consumption showed the opposite pattern, with most chicks consuming all of the fish available. However, cat food consumption was quite high considering that cat food was less frequently the first or second choice of the chicks, compared to the marine foods. The amount of cat food consumed appeared to decrease with increasing time spent in captivity, while mussel consumption varied widely between individuals and across test days (Fig. 3).We found no significant differences between the chicks in their first and second food choices, nor

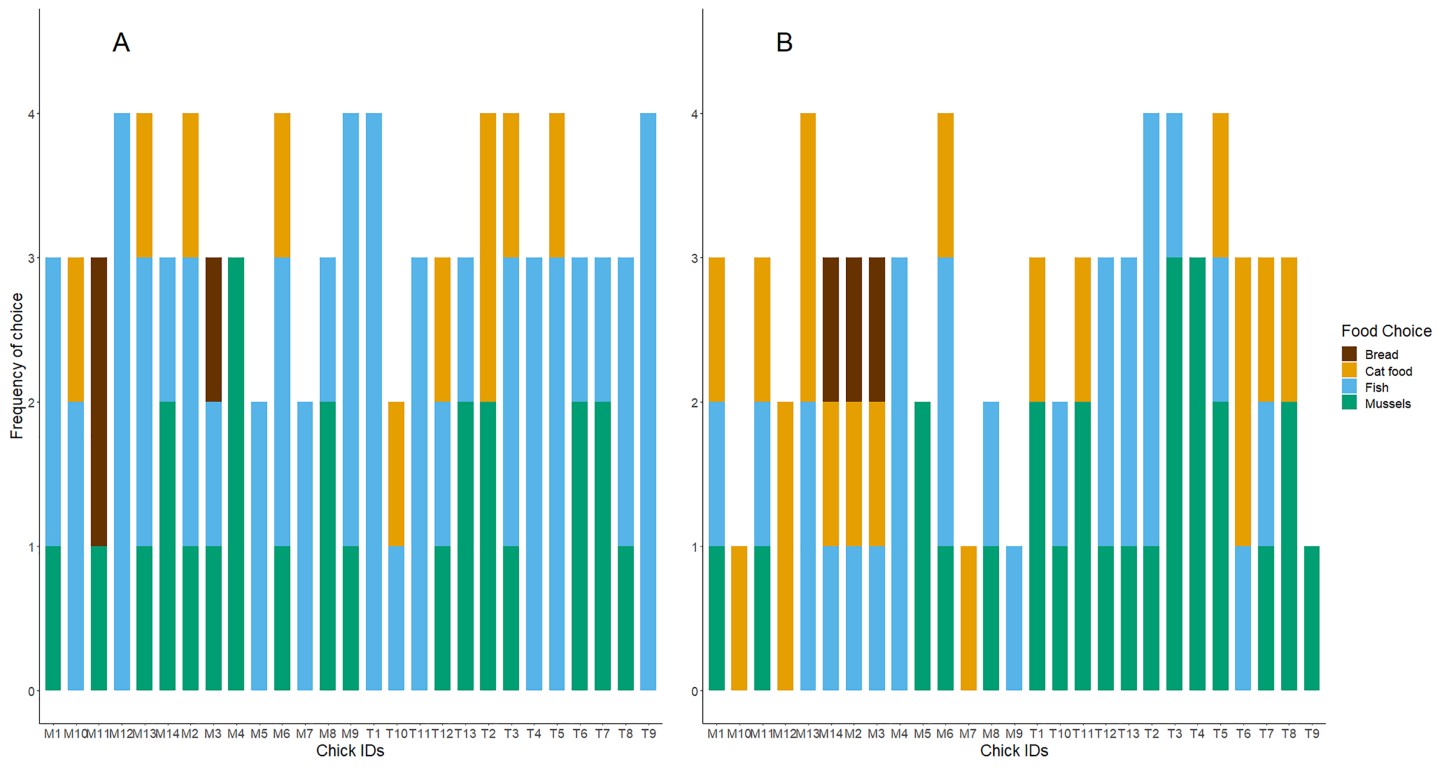

**Figure 2 The frequency of foods consumed as first and second choices for each test subject.** (A) Shows the first food choice each chick made, and (B) shows the second food choice. The x-axis displays each chick's unique study code. The height of each stacked bar shows the cumulative sum of each chick's participation and consumption of food in the four food preference trials. Bar colours refer to food types.

significant within-individual consistency between test trial days 5, 10 and 15; day 35 could not be included in this analysis due to reduced chick participation (Table S2 in the Supplemental information; all ICCs: $P > 0.05$).

## Do chicks prefer higher protein foods over lower protein foods?

Higher protein foods were generally more frequently consumed as chicks' first choice (Fig. 4A) and second choice (Fig. 4B) during the food preference trials. However, exact binomial tests of a high protein food being chosen first (yes or no, probability by chance = 0.5) showed that only in the day 5 trials chicks showed a significant preference for high protein foods for the first food consumed (Table 3). Chicks generally ate higher percentages (>50%) of high protein food first in trials on days 10, 15 and 35, but this was not significantly different from the 50% expected by chance. Within the first two food choices made during trials, almost all chicks chose a high protein food (Table 3). Exact binomial tests of the frequencies of choice for higher protein foods within the first two choices suggest that chicks preferentially chose higher protein foods on days 5 and 10 ($P = 0.02$). Yet, their first two choices for high-protein foods on days 15 and 35 did not differ significantly from chance ($P = 0.11$ and $P = 0.71$ respectively; Table 3).
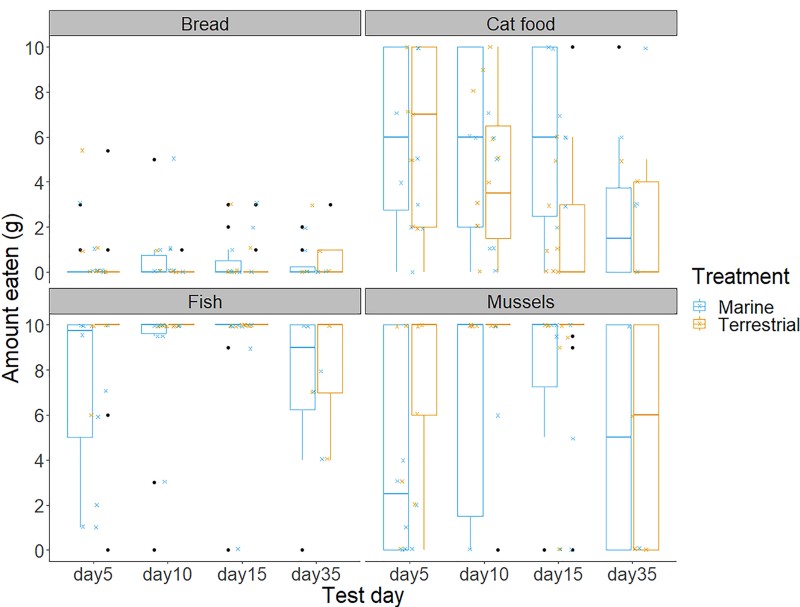

**Figure 3 The amount of bread, cat food, fish and mussels consumed by the chicks in the 10 min food preference trials on each of the test days.** Boxplot outline colours represent the diet treatment groups (blue for Marine and orange for Terrestrial) on each test day. Each box represents the interquartile range of the amount of food eaten (grams), with the solid horizontal lines representing the median values. Values outside of the interquartile range are represented by the vertical lines and black dots represent extreme values. The smaller jittered crosses show the raw data for each of the 88 observations, and colour indicates the diet treatment group to which each individual was assigned.

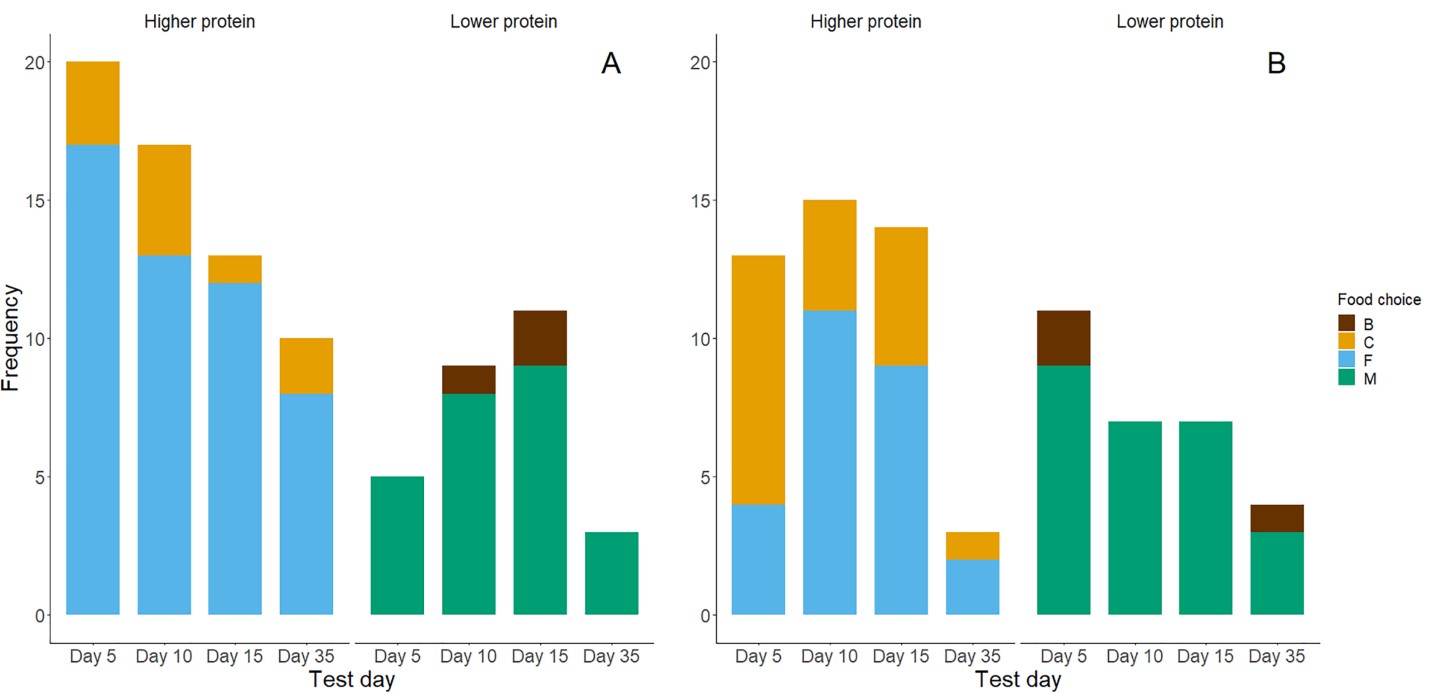

**Figure 4 Bar graphs of chicks' first (A) and second (B) food choices in each trial, according to the food types' protein levels (high vs. low).** The bar colour indicates whether a higher protein food was chosen (yes or no). (A) Represents $n = 88$ observations where chicks made a first food choice and (B) represents $n = 74$ observations where chicks made a second food choice.

**Table 3 Exact binomial tests of probabilities for higher protein food consumed first (0.5) and for whether consumed within the first two foods consumed by chicks (0.83) across the test trials.**

| Test day | Frequency of higher protein food consumed first | P-value for first consumed food | Number of chicks that participated (consumed food) | Frequency of higher protein food consumed within the first & second choices | P-value for first & second food consumed |
|---|---|---|---|---|---|
| Day 5 | **20** | **<0.001** | **25** | **25** | **0.015** |
| Day 10 | 17 | 0.169 | **26** | **26** | **0.015** |
| Day 15 | 13 | 0.839 | 24 | 23 | 0.107 |
| Day 35 | 10 | 0.092 | 13 | 12 | 0.710 |

Note:
Significant (α < 0.05) stats are highlighted in bold.

## DISCUSSION

We tested whether rescued herring gull chicks had individual food preferences, and whether those preferences were influenced by their rearing diet in captivity. We found that herring gull chicks unequivocally preferred marine food, specifically fish, and avoided the bread offered. Individual preferences for cat food and mussels were not clear nor consistent, but mussels were picked more often as a first or second option than cat food.

Recently-hatched chicks are usually provisioned by the parent gull with pre-digested, mixed regurgitates on the floor and presented with smaller, appropriately sized, amounts of food by the parent directly into the chick's beak (*Tinbergen & Perdeck, 1950*). For the first few days post-hatching, chick vision and acuity are not fully developed, so visual identification of food may be beyond chick abilities until they are a few days older and more developed (*Segovia et al., 2020*; *Thompson, 1971*). This may contribute to the lack of preference in our baseline food preference trial with younger chicks, as there may not be a biological imperative to have preferences until they can distinguish the prey provided. However, from the first 2×2 food preference test onwards, virtually all chicks preferred fish, regardless of their experimental rearing diet. Marine-based diets appear to facilitate gull fledgling productivity in terms of growth and survival (*Spaans, 1971*; *Bukacińska, Bukaciński & Spaans, 1996*; *Annett & Pierotti, 1999*; *Sotillo et al., 2019*). Marine prey, such as pelagic fish, are thought to contain better nutrition (protein and micronutrients) for chick development than terrestrial anthropogenic foods like domestic refuse (*Pierotti & Annett, 1991*; *Nogales, Zonfrillo & Monaghan, 1995*; *Bukacińska, Bukaciński & Spaans, 1996*; *O'Hanlon, McGill & Nager, 2017*; *Sotillo et al., 2019*). Calcium, for example, is one of the essential micronutrients required for skeletal growth, and the calcium in fish bones is easily digested and absorbed by growing gull chicks (*Spaans, 1971*; *Annett & Pierotti, 1989*; *Pierotti & Annett, 1991*).

We did not find any evidence of distinct among-individual preferences or within-individual consistency in food choices. We also did not find a strong preference for the higher-protein food options, contrary to expectations. These results are likely to have been driven by the chicks' strong preference for fish and avoidance of bread. Our findings
on herring gull chicks' unwillingness to consume bread appear to be at odds with a study on wild, urban black-headed gulls (*Chroicocephalus ridibundus*) which found that gulls in more urban areas preferred bread over fish, while gulls in rural settings exhibited the opposite preference (*Scott, Duncan & Green, 2015*). An increased reliance on anthropogenic foods such as bread may emerge later in life, perhaps when older individuals experience lower foraging returns from attempts to feed on dwindling marine prey (*Greig, Coulson & Monaghan, 1986*; *Bicknell et al., 2013*; *Oro et al., 2013*; *Blight et al., 2015*; *Carmona, Aymí & Navarro, 2021*; *Pais de Faria et al., 2021*). In addition, gull chicks and independent juveniles have also been observed to consume bread in the wild, which may be due to the reliability and ease of access to anthropogenic foods for the parent gulls to forage (*Spaans, 1971*; *van Donk et al., 2017*; *Sotillo et al., 2019*). In addition, juveniles may consume less nutritious foods, such as bread in parks, due to competition and exclusion from higher-quality resources by adult gulls (*Verbeek, 1977*; *Greig, Coulson & Monaghan, 1983*; *Carmona, Aymí & Navarro, 2021*; *Pais de Faria et al., 2021*). Further research is required to unravel how food preferences and foraging specialisations develop and change through individuals' lifespans. Willingness to consume novel anthropogenic foods has been shown to vary in birds across urban-rural gradients (*Blight et al., 2015*; *Brousseau, Lefebvre & Giroux, 1996*; *De León et al., 2019*; *Langley et al., 2021*; *Scott, Duncan & Green, 2015*). However, for many animals, the consumption of anthropogenic foods could be an evolutionary trap, as described by the 'junk-food hypothesis', where individuals shift from consuming scarce, high-quality foods to consuming more abundant but nutritionally lower-quality foods (*Auman, Meathrel & Richardson, 2008*; *Grémillet et al., 2008*; *Sol et al., 2014*; *Stillfried et al., 2017*; *De León et al., 2019*). Whether herring gulls suffer later-life detrimental effects on their health when they increasingly rely on human food waste, remains to be determined, however.

We did not find any influence of early-life experimental diets on later food preferences, although the chicks appeared to gain a predominant and enduring preference for fish and avoidance of bread in virtually all tests after the initial baseline test. Individual experience and learning in early life are thought to underlie young seabirds' acquisition of foraging specialisms in later life that might help escape conspecific competition (*Anderson et al., 2009*; *Borrmann et al., 2021*; *Patrick et al., 2014*; *Votier et al., 2017*). Early-life experience through social learning or maternal effects has been shown to influence food preference and foraging specialisation. Weanling rats (*Rattus norvegicus*), for example, are more willing to consume food with onion flavouring if they experience this taste from their mother's milk (*Wuensch, 1978*). Bottlenose dolphins (*Tursiops aduncus*) and other odontocetes show later-life specialisms in prey choice and foraging behaviour from learning in early life from their mothers and other group members (*Allen, 2019*; *Strickland et al., 2021*). Individual or group variation in prey choice allows multiple ecotypes to coexist with reduced competition. Knowledgeable individuals can either directly or indirectly pass information on to juveniles as to which foods to consume (*Monaghan, 2007*). Our experiment did not test chicks' food preferences in a social context (such as sibling competition), nor did our test subjects have any access to a more knowledgeable individual (*e.g.*, their parents as in wild juveniles) that could have increased individual

variation in food preferences, as observed in wild primates (*Brown, Almond & Bergen, 2004*), white-tailed ptarmigan (*Lagopus leucurus*, *Allen & Clarke, 2005*), meerkats (*Suricata suricatta*, *Thornton & Malapert, 2009*) and New Caledonian crows (*Corvus moneduloides*, *Holzhaider, Hunt & Gray, 2010*). As herring gull chicks may refuse to consume certain foods, as seen in our study, this may reinforce provisioning behaviour seen in the wild where parents provide more marine foods to their chicks (*Spaans, 1971*; *Annett & Pierotti, 1989*; *Pierotti & Annett, 1991*; *Nogales, Zonfrillo & Monaghan, 1995*; *van Donk et al., 2017*).

Our diet treatments were designed to replicate extremes in the ratios of marine and terrestrial foods that wild gulls have been recorded to provide their chicks with (*Pierotti & Annett, 1991*; *Sotillo et al., 2019*). Though we presented the different foods for differing amounts of time to make the dietary ratios, all foods were made available to all of the chicks, which may not be the case in the wild (*Brower, Spaans & De Wit, 1995*; *Duhem et al., 2003*; *Mendes et al., 2018*; *Pierotti & Annett, 1991*; *van den Bosch et al., 2019*). We found that our terrestrial diet group chicks were lighter and their tarsus measurements were smaller prior to release (Figs. S1 and S2). Our findings differ from studies on captive lesser black-backed gull chicks, where those reared on chicken breast grew faster and were larger in mass at 30 days old than those that ate more fish (*Gupta et al., 2016*; *Sotillo et al., 2019*). Anthropogenic foods of terrestrial origin, like household food waste (*e.g.*, animal remains), can be high in calories and protein, which would meet chick requirements for growth and development (*Gupta et al., 2016*; *Sotillo et al., 2019*; *van Donk et al., 2017*; *van der Meer et al., 2020*). Our terrestrial diet group chicks may have been lighter because, once they were >25 days old, they started to refuse the terrestrial foods provided for the majority of the time (with time available used to create the experimental diet ratios). Instead of eating the terrestrial foods, they often waited until the marine foods were provided (as per the diet time ratio). All food was provided *ad lib* and all chick weights were within the natural, healthy ranges at last measurement (*Spaans, 1971*). However, to avoid reduced daily food intake, we provided marine and terrestrial foods simultaneously for the remainder of the gull chicks' time in captivity. The refusal to consume bread and to some extent, cat food, may point to dietary conservatism (*Marples & Kelly, 1999*). Although we could not distinguish variation in individuals' dietary conservatism nor test for social competitive effects, such factors could influence individual food preferences in later life.

This study shows that investigating the development of individual food preferences of animals is important to understand and predict how species may cope with increasing urbanisation and climate change. Animals can live in and exploit urban areas for anthropogenic food waste and refuse. However, this does not necessarily mean that they are thriving or that they prefer anthropogenic food, rather than making the best of a bad situation (*Oro et al., 2013*; *Soulsbury & White, 2015*). Further research on the consequences of early-life provisioning of anthropogenic foods on later-life food and habitat preferences is needed on a wider range of taxa. It would also be of interest to study knock-on effects of food preferences and diet on longevity, reproductive success and demography.

## ACKNOWLEDGEMENTS

We would like to thank Suzy Sharpe for her experienced care and guidance, and for the use of her facilities to help rear these gull chicks to release and, without whom, this study could not have been conducted. We thank the ornithological specialist vets, in particular Dr Felicity Woodhouse, who advised and helped to oversee the care of the juvenile herring gulls in this study. Also, thanks to Shubhi Raghav for help with coordinating tests on a couple of very busy mornings. We would also like to thank our reviewers Dr Lamarre, Dr Stienen and a third anonymous reviewer for their invaluable comments that have contributed to making this a much better manuscript.

### Funding

Emma Inzani is funded by a NERC GW4+ PhD studentship NE/S007504/1 and NE/L002434/1. Neeltje J. Boogert and Laura Kelley are funded by Royal Society Dorothy Hodgkin Research Fellowships (NB: DH140080, LK: DH160082). The funders had no role in study design, data collection and analysis, decision to publish, or preparation of the manuscript.

### Grant Disclosures

The following grant information was disclosed by the authors:
NERC GW4+PhD Studentship: NE/S007504/1 and NE/L002434/1.
Royal Society Dorothy Hodgkin Research Fellowships: DH140080, DH160082.

### Author Contributions

- Emma Inzani conceived and designed the experiments, performed the experiments, analyzed the data, prepared figures and/or tables, authored or reviewed drafts of the article, and approved the final draft.
- Laura Kelley conceived and designed the experiments, analyzed the data, authored or reviewed drafts of the article, and approved the final draft.
- Robert Thomas analyzed the data, authored or reviewed drafts of the article, and approved the final draft.
- Neeltje J. Boogert conceived and designed the experiments, analyzed the data, authored or reviewed drafts of the article, and approved the final draft.

### Animal Ethics

The following information was supplied relating to ethical approvals (*i.e.*, approving body and any reference numbers):

This study was approved by the University of Exeter, College of Life and Environmental Science, Penryn Ethics Committee (eCORN002962 9.1). GPS tagging and marking of chicks with leg rings and non-toxic temporary dye were approved by the British Trust for Ornithology Special Methods Technical Panel on behalf of the Joint Nature Conservation Committee & Natural England (application licence numbers: 11962 and 11963).

## Data Availability

The raw data and code are available in the Supplemental Files.

## Supplemental Information

Supplemental information for this article can be found online at http://dx.doi.org/10.7717/peerj.17565#supplemental-information.

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
