# Peer review of "Early-life diet does not affect preference for fish in herring gulls (Larus argentatus)"

_PeerJ, doi:10.7717/peerj.17565_

## Round 0.1 · original submission · Major Revisions

Dear Authors,

After receiving comments from three reviewers, all agree that the manuscript is interesting but still needs major corrections before it can be accepted for publication. Mainly, it is necessary to better explain the experimental design, better structure the discussion based on its hypotheses and objectives, as well as correct several grammatical errors.

Best regards,

Armando Sunny

**Language Note:** The Academic Editor has identified that the English language must be improved. PeerJ can provide language editing services - please contact us at [email protected] for pricing (be sure to provide your manuscript number and title). Alternatively, you should make your own arrangements to improve the language quality and provide details in your response letter. – PeerJ Staff

·

Excellent Review

This review has been rated excellent by staff (in the top 15% of reviews)
EDITOR COMMENT
The feedback provided consistently highlights crucial areas for enhancement within the manuscript, focusing on essential points that require attention in both the introduction and the Experimental design. The reviewer's comments are remarkably clear, precise, and timely. This thorough and insightful revision represents an outstanding effort that promises to significantly elevate the quality of the manuscript. The reviewer deserves commendation for their invaluable contribution, and their dedication to refining the work is truly praiseworthy. Sincerely, Armando Sunny

Basic reporting

This study presents a very interesting premise where rescued gull chicks are exposed to a marine or terrestrial diet during early development and then tested on their preference for terrestrial or marine foods repeatedly during the nestling stage. The authors could then test whether the chicks’ preference for certain food items related to the diet they regularly received or whether certain items or nutrient profiles were favoured outside of the chicks’ experimental diet. They found that all gull chicks consistently preferred fish and avoided bread, regardless of dietary treatment.

While I believe that this study makes a significant contribution towards our understanding of inter-individual variation in foraging strategy within a species known to exploit a wide range of resources, I have some concerns regarding the context within which this study is presented, the information missing from the methodological and result sections, and the misuse of several references. I provide broad descriptions of my main concerns below followed by line-specific comments.

One important piece of information missing from the introduction might be to explain what makes anthropogenic food resources to be considered low quality. Anthropogenic food is equally described as “junk-food” low nutritional quality, low energetical quality; all terms used to describe fishery discards as much as urban food waste. Shifting one’s diet from diving for pelagic fish to exploiting fishery discards is very different compared to being a terrestrial omnivore supposed to eat berries and insects but now only relying on processed human food. Similarly, paragraph two described the shift to an anthropogenic diet as inducing both hyperglycemia (in raccoons) as well as low body condition (in gannets); these two effects are contradictory because the types of anthropogenic resources that induce these effects are completely different. I agree with the authors that both sides of foraging on anthropogenic resources are important to discuss but I would argue that they need to be discussed separately; on the one side, what happens when highly-specialized foragers now have access to fishery discards of lower energy/macronutrient density. On the other side, what happens when generalist species gain access to energy-dense food that might or might not contain certain key nutrients (e.g. taurine, omega-3s, fibre, etc). Both types of diet might very well lead to malnutrition but in the case of specialized seabirds, it might reduce their fitness in the short term whereas for urban generalists, it might actually increase their fitness (at least on the short-term).

The review of the literature at paragraph 3 on the positive effects of consuming anthropogenic resources lacks thoroughness and conciseness. The examples given lack cohesiveness as we go from injured adults to the body condition of seemingly healthy individuals, to pros and cons of reliable food sources for birds to survive the winter. Since your study is all about chick provisioning, it would been important to include information about how reliance on anthropogenic food sources influences breeding success. Many avian studies have been done on the subject with many finding positive associations between markers of breeding success (egg/clutch size, chick weight, fledgling rate, overall population growth) and reliance on landfills or human-made food. Some examples include:

Duhem et al. 2008. Effects of anthropogenic food resources on yellow-legged gull colony size on Mediterranean islands. Population Ecology 50: 91–100

Serré et al. 2022. Lake Superior herring gulls benefit from anthropogenic food subsidies in a prey–impoverished aquatic environment. Journal of Great Lakes Research 48: 1258-1269

Steigerwald, et al. 2015. Effects of decreased anthropogenic food availability on an opportunistic gull: evidence for a size-mediated response in breeding females Ibis 157: 439-448

Weiser and Powell. 2010. Does Garbage in the Diet Improve Reproductive Output of Glaucous Gulls? The Condor 112: 530–538

Furthermore, there are several instances of mis-referencing throughout the manuscript, but especially in the discussion. I detailed each instance found in the additional comments below but a throughout review of the manuscript by the authors is required to ensure that each statement is referenced by the appropriate citation and that the findings from previous studies are not being misrepresented.

Experimental design

All the information regarding the origin of the subjects is missing. This information is critical to put the experimental design in a biological context. Could the chicks have originated from remote colonies with only access to marine resources prior to being brought into captivity? Or could they have only come from urban nesting sites where they would have had access only to garbage? Perhaps their parents had equal access to marine and anthropogenic resources? This information would inform us about the possible content of their eggs and what their parents might have fed them prior to their admission to the rehab facility.

Although the data on growth were collected throughout the experiment, they are not included in the results as a function of dietary treatment groups. This seems like a missed opportunity to further understand the impact of a so-called junk food diet on the growth of chicks throughout their development. Furthermore, a large chunk of the discussion is dedicated to discussing the effects of garbage intake on growth rate – this is only relevant if previous findings can be compared to this study’s findings.

Validity of the findings

There are a few instances where the results are overstated, particularly in how they provide information about social and environmental components of HERG’s ecology.

There was also a missed opportunity to look at individual consistency in food preference, which was never quantified despite certain statements in the discussion claiming otherwise. This piece of information could be included, however, since the data exist to quantify individual consistency across trials.

Lastly, the relationship between preference for marine vs terrestrial or high-protein vs low-protein is not well-disentangled such that I am not convinced that the chicks’ “preference” for high-protein options is not simply a strong preference for fish coupled with an aversion to bread. I suggest below additional analyses and data visualization to ensure that the high-protein preference result is not blurred by any specific food item.

Please find my specific concerns associated with line numbering in the additional comment section.

Additional comments

Please find specific comments associated with line numbers below.

Abstract:
Lines 40-41: There is conflicting evidence regarding the statement that “most” breeding gulls shift towards a marine diet to feed their chicks. Many studies have found gull parents from different species to maintain a diet high in garbage throughout the breeding season or even shift their diet towards exploiting more landfills:
e.g. Lenzi et al. 2019. The impact of anthropogenic food subsidies on a generalist seabird during nestling growth. Science of the total environment 687: 546-553.

Steigerwald, et al. 2015. Effects of decreased anthropogenic food availability on an opportunistic gull: evidence for a size-mediated response in breeding females Ibis 157: 439-448

Weiser and Powell. 2010. Does Garbage in the Diet Improve Reproductive Output of Glaucous Gulls? The Condor 112: 530–538

I would suggest changing “most” to “many” to be more accurate


Line 47: You report a final sample size of 30 here but the results section reports 27 subjects; discrepancy between the two numbers provided.


Line 51: Unclear what “food availability” refers to here since food was given to chicks ad lib.


Lines 52-53: The wording here seems to suggest that parents shifted their foraging habits towards exploiting marine resources due to chick preference but that happens even before the chicks could show a preference (i.e. upon hatching)? It’s likely more accurate to describe chicks’ food preference as matching the parents’ natural tendency to shift to a marine diet because it’s the most nutritious during development


Intro:
Lines 64-65: This statement is quite vague, it would be useful to state whether this is referring to resources that are simply absent from urban environments without similar replacements: i.e. flying insects for prey, marsh-like habitats or whether this statement refers to natural resources that are somewhat emulated by anthropogenic resources: e.g. tall buildings as trees, attics/chimney/bridges as caves, etc.


Lines 76-81: All of the examples stated above pertain to young animals. It would be important to mention that it’s during the early-life stage that these negative outcomes are perceived; there is currently not much evidence suggesting that adults suffer negatively from foraging on fishery discards (or other type of anthropogenic resources) but there is evidence that it matters during pregnancy/egg-laying + lactation/chick-provisioning stages.


Lines 85-88: This sentence seems to imply that eating a lower quality diet causes more injuries whereas all the papers cited here report that it’s the animals’ proximity to humans that might led to more injuries (i.e. hit by car, intentional trapping, or ingesting harmful material (i.e. plastic, PFAS, etc) - the distinction between both is important to make.


Lines 85-88: It’s unclear which ones of the papers cited here report that high intake of anthropogenic resources causes poorer body condition and higher parasitic load - Many of these papers suggest that diseased/weakened/injured animals make more use of urban resources but they link that to lower competition and inability to hunt. While we cannot exclude that feeding on anthropogenic resources might lead to these issues, those were not the findings of these studies but rather that poorer-quality animals end up having to rely on human-made resources whereas higher-quality individuals could retain their natural range and diet.


Lines 114-128: It is unclear what new information paragraph 4 is conveying. Several seabird species have been discussed above and it’s unclear how the information about gulls entering into conflicts with humans fits with the rest of the study. Since the introduction is fairly long, I would suggest cutting this paragraph out.


Lines 117-119: This sentence would benefit from adding more details regarding what the “natural marine prey” referred to (pelagic fish presumably?) and what are the direct and indirect threats that we pose to them - fishing, oiling, pollution, ocean warming?


Lines 130-132: This statement should be referenced and perhaps nuanced as some studies have found inter-season variation in foraging habits within individuals, e.g.:

Davis et al. 2017. The glaucous-winged gull (larus glaucescens) as an indicator of chemical contaminants in the canadian pacific marine environment: Evidence from stable isotopes. Archives of Environmental Contamination and Toxicology, 73(2), 247-255.

Peterson et al. 2017. Mercury contamination and stable isotopes reveal variability in foraging ecology of generalist California gulls. Ecological indicators, 74, 205-215.

Rolanda et al. 2011. Seasonal and Age-Dependent Dietary Partitioning between the Great Black-Backed and Herring Gulls, The Condor 113: 795–805


Lines 130-136: It would be worth pointing out that certain individuals of certain species are much more consistent in their foraging strategy across seasons (e.g. GBBG) whereas other species/population seems to show a lot more inter-season variability (YLGU)


Lines 130-140: This paragraph seems slightly out of place. I would suggest moving the information regarding gulls in the next paragraph before discussing HERG chicks.


Line 142: It would be beneficial to include a definition of foraging strategy here. In the context of this study, does it pertain to distinct types of extracting strategy (i.e. pecking vs diving vs digging), to the types of environment exploited (i.e. pelagic vs underwater vs forest vs marsh vs landfills) or the types of food exploited (i.e. garbage in bins vs fish vs plants vs mammals)?


Line 142: Does maternal effect only apply to mammals, or does it apply to the egg content as well? It would be important to distinguish the two as the intro goes back and forth between discussing birds and mammals.


Lines 143-146: It would be important to mention whether we know if gull chicks exhibit food preference at the nest when competing with their siblings. Per experience, it seems like chicks are in such a rush to consume the largest amount of food possible when fed by their parents that the type of food given to them at the moment is not relevant.


Lines 142-156: This paragraph would benefit from being restructured as, at this point, herring gulls haven’t been introduced as the study’s subjects, therefore it seems out of place to point them out specifically when the rest of the intro has been about mammals and birds in general.


Lines 162-163: The Sotillo paper is about LBBG - it would be more appropriate to show the dietary range within the HERG species. Many papers from HERG nesting around the Great Lakes in Canada show their propensity to rely on an anthropogenic diet during the breeding season, e.g.:
Serré et al. 2022. Lake Superior herring gulls benefit from anthropogenic food subsidies in a prey–impoverished aquatic environment. Journal of Great Lakes Research 48: 1258-1269.


Lines 165-166: If the statement regarding breeding gulls switching to a marine diet to provision their chicks pertains solely to HERG, it should be indicated as such. Otherwise, many studies have found gull parents from different species to maintain a diet high in garbage throughout the breeding season or even shift their diet towards exploiting more landfills:
e.g. Lenzi et al. 2019. The impact of anthropogenic food subsidies on a generalist seabird during nestling growth. Science of the total environment 687: 546-553.

Steigerwald, et al. 2015. Effects of decreased anthropogenic food availability on an opportunistic gull: evidence for a size-mediated response in breeding females Ibis 157: 439-448

Weiser and Powell. 2010. Does Garbage in the Diet Improve Reproductive Output of Glaucous Gulls? The Condor 112: 530–538


Lines 167-169: It’s unclear what this alternative hypothesis indicates: would the junk-food hypothesis predict a preference based on habituation in young chicks? In adults, foraging on anthropogenic resources seems to be favoured either because garbage is highly reliable in location/quantity or because it requires less energy to exploit. More details are needed.


Line 171: There was no background information regarding preference for food high in protein across generalists previously in the intro - what is the rationale behind this hypothesis?

Methods:
Lines 185: Information missing from this paragraph regarding where these birds could have come from - is the rehab centre located in a city such that the chicks could have only come from urban nesting sites? Or could they have come from colonies with both access to marine and anthropogenic resources? This is an important piece of information to have regarding the possible content of their egg and what their parents might have fed them before they were brought into rehab.


Line 250-251: Same comment as above regarding the Sotillo paper not being about HERG and instead, using a study specific to HERG to show their large food range.


Lines 253-254: It’s unclear whether was a methodological decision to use a plant-based product (bread) as the terrestrial low-protein but use an animal product for the marine low-protein option (mussel). Because their nutrient profile differs in so much more than just in protein (i.e. ratio carb:lipid is completely different between these two options), more details are needed to explain the rationale behind this methodological choice. Presumably, comparing bread with a marine equivalent (e.g. food item made of seaweed) would have been considered?


Lines 263-264: Were the food items leftover at each meal change quantified as another data point regarding food preference, i.e. if across all groups, bread was always left over vs fish was never leftover?


Line 305: Specify that this is the baseline food preference test (instead of “first experimental test”) as this was unclear upon first reading this paragraph.


Lines 305-307: Since it was mentioned above that very young chicks were fed by tweezers, was it ever the case that chicks might have been too young to properly participate in this baseline food preference test because of poor mobility or unability to self-feed? Or were those excluded from the study if they were still being hand-fed past 72hrs upon having been brought into captivity?


Lines 321-322: How were chicks assigned to their treatment group: randomly assigned, block randomization? Of the chicks that initially showed a preference for fish, how many ended up in the marine group vs in the terrestrial group?


Lines 322-325: This information should be moved to the end of the next paragraph as it pertains to the actual food preference trial, not the baseline test.


Lines 328-329: A figure showing the experimental design or, even better, a video showing a chick undertaking the food preference test would be helpful to include in the manuscript.


Lines 366-369: Looking at figure 2, it seems like whatever response you will get here will be mainly driven by the gulls’ aversion to bread + their preference towards fish – at this point, I am not convinced that these analyses show a preference for high protein food rather than an overall preference for 1 high-protein food coupled with an avoidance of 1 low-protein food. To disentangle this effect, it might be worth repeating the chi-squared tests from lines 364-366 but for the second food choice. If the second food choice comes out consistently as cat food, then you can be sure that chicks prioritize high-protein food first. If it comes out as a mix of mussel and cat food, then there might be consistent individual variations making some chicks favour marine food all around while others favour high-protein food. If this is the case, an analysis of individual consistency across trials would be very interesting to perform to see whether food preferences beyond fish are carried through throughout the nestling stage.


Results:
Lines 377-379: It would be important to include the analysis regarding treatment group vs food preference in the result section (not just in the supplementary material) as this seems to be the main finding of your paper regarding individual preference vs habituation.


Lines 388-389: It’s unclear where the output of this GLMM is and what the subject variation captured by the random effect is.


Lines 387-391: This section only described inter-individual variability but what about intra-individual repeatability? It would be important to include statistics to show whether individuals were consistent in their food preference across trials.


Lines 398-399: This set of results is missing context to better visualize the results - the no preference for high or low-protein seems to arise because gulls ended up just preferring the two types of marine food presented vs the cat food and the bread. See my comment at Figure 4 to better demonstrate these relationships.


Discussion:
Line 408: Individual food preference was never assessed as within-subject consistency across trials was never assessed. I would suggest adding this analysis, as per my comment above.


Line 408: It’s incorrect to describe the subjects of this study as representing a population since we do not know whether they all come from the same colony or whether they had other markers that would make their origin homogenous. I would suggest rewording “population-level” as “group-level”.


Lines 410-411: It does seem like you have the data to show that chicks ranked their food preference in the following order: 1-fish, 2-mussel, 3-cat food, 4-bread (based on Table 1). As mentioned above, more analyses are required to show whether HERG prioritized marine food ahead of terrestrial food, even when the protein content of the terrestrial food might have been higher.


Lines 410-411: Not consistent at the individual level or at the group level?


Lines 411-414: Several instances of mis-referencing:
Nogales et al. 1995; Ronconi et al. 2014 did not study fledgling rate. Pierrotti and Annett 1991 found that diet had no effect on fledging success. The Sottillo et al. 2019 study did find that a terrestrial diet was associated with slower growth rate in the wild but could not replicate those results in captivity.


Lines 418-421: Several instances of mis-referencing:
The references included here are not the primary source of information for the statements made regarding 1) the bioavailability of calcium in fish bone, 2) the lack of calcium in crops or man-made food like bread, and 3) gulls not being able to digest carbs properly. Please use only the primary references associated with each one of those statements.


Lines 433-435: It’s unclear why urban nesters are described here as being more restricted in their movements? All 3 studies cited here found that gulls not specializing on fish had much larger foraging ranges.


Lines 435-436: This statement is quite vague, it should be clear that you are implying here that some urban nesters might shift to a more marine diet outside of the breeding season.


Lines 435-437: Instances of mis-referencing:
O’Hanlon et al. 2022 did not compare the gulls’ diet when breeding vs not breeding. They only report their diet outside of the breeding season.
Shamoun-Baranes et al. 2017 did not assess the birds’ diet and actually showed a wide range of variation between adults remaining inland throughout the years and others that showed some bouts at sea. Both of these references do not support the statement made here.


Lines 437-439: This statement about the diet of wild boar seems out of place. Your study is all about marine vs anthropogenic food in a migratory species, it is a stretch to compare it to an exclusively terrestrial omnivore.


Lines 444-445: Specify that this pertains to herring gull chicks only, it’s never been tested in adults.


Lines 449-451: As mentioned above, I am not yet convinced that your data support this interpretation. I think that the few instances where preference for high protein food came out as significant was solely driven by the gulls’ propensity to eat fish first. As mentioned above, more analyses are required to disentangle marine/terrestrial and high-protein/low-protein preferences.


Lines 451-453: This is likely an overstatement (RE: preference for high-protein must be adaptative) not supported by the data as it currently stands.


Lines 455-456: Since you have the gulls’ morphometric data, I would suggest running a comparative analysis between the growth curve of both treatment dietary group (terrestrial vs marine) in order to compare your results properly with previous studies’ findings.


Lines 461-483: This full paragraph on the effect of anthropogenic food resources on chicks’ growth is out of place without being able to compare your own results to those of the studies mentioned here. As mentioned above, the growth curve of both dietary treatment groups should be compared and included in the study’s main results.


Lines 470-472: Those two explanations seem to contradict each other - what findings cited here imply that change in body size due to urbanization is a maladaptive response?


Lines 461-483: There is a lot of contrasting information in this paragraph that would benefit from being ironed out. The literature underlying the negative effects of an anthropogenic diet on growth is quite compelling and could be directly compared to your morphometric data. More subtle consequences of a junk-food diet like increased fat deposits seem less relevant to discuss here and no parallel can be drawn between this information and your study. I would suggest sticking only to parallels that can be drawn between your data and the previous literature on growth rate.


Line 494-495: This statement (RE ecotypes) is very interesting and should be properly referenced.


Lines 506-510: It’s unclear here why garbage containing undigestible material is being discussed here since this was not an experience that the chicks had in captivity and, based on your next statements, they would have been unlikely to encounter this issue prior to being brought into captivity given their overall young age. This information seems to be irrelevant to the context of this study.


Lines 513-515: A piece of information that appears to be missing from the manuscript is how HERG parents deliver food to their chicks. Per personal observation, it seems like chicks will peck their parents’ bill, the parents will regurgitate, and the chicks will eat both by reaching directly into the bill of their parents and by eating the food regurgitated on the ground. There do not seem to ever be pauses in feeding events where the chicks will just look at the food or poke at it, they just seem to ingest as much of the regurgitate as fast as possible. Of course, the conditions are completely different in captivity where a lack of competition, particularly during the food preference test, can now show individual preferences, but in an ecological context, can the chicks’ individual food preferences ever really be shown to the parents? And in return, would the parents ever adjust their foraging strategy to match their chicks’ preferences? This appears ecologically unlikely unless the chicks’ preferences align with a diet that enhances their fitness at a population level. This information should be discussed here to place your findings within a biological/ecological context.


Lines 517-520: It’s unclear here whether you are suggesting that because young chicks can only manipulate certain food (soft, small pieces), this will influence their food preference later on? Otherwise, it’s unclear what physical limitations and food preferences have in common here since those physical limitations will be irrelevant as the chicks grow.


Lines 520-524: This appears to be an overstatement since the sentence below suggests that all food options were similarly easy to manipulate and digest. But is that so? It seems like pieces of whole fish might be harder to digest given the presence of bones and scales compared to very soft food like cat food or mussels? Alternatively, were small chicks able to manipulate perhaps slippery chunks of cat food as easily as pieces of fish? A video showing the chicks’ manipulation of each food option would be helpful. At the very least, a picture of each food option + the experimental setup would be required for the reader to better visualize the methods used here.


Lines 528-534: None of the references cited here studied chicks preference or avoidance of certain foods and how that might influence the parents’ behaviour - instead, they suggest that when parents shifted their foraging habits towards a more fish-heavy diet, this shift occurred upon chick hatching - not in response to chicks’ preference.


Line 541-544: Neither of the references cited here studied or showed a relationship between begging and a change in the type of food brought back by the parents.


Lines 546-550: It would be more appropriate to follow each trait brought up with the reference it came from rather than listing this long list of traits and including the references at the end; this almost suggests that all of these references studied each one of these traits and came to the same conclusions across the board.


Lines 550-553: Is this true for gulls or has that only been tested in other species? It would be helpful to specify within which taxonomic group this has been observed.


Line 557: Common name associated with the latin name of pheasant species is missing.


Lines 572-574: It’s unclear which part of the study included a social and environmental component as discussed here - the origin of these chicks was either unknown or not considered and the food preference tests were presented to individual chicks such that any social component (competition or social learning) was removed.
Also, it’s an overstatement to describe the findings as predictive of how species might deal with climate change - presumably if the growth and final body condition of chicks from both terrestrial and marine diet groups did not differ from each other, that implies that the terrestrial food was eaten in quantities that provided similar nutrition as the marine food. So, we might instead conclude that when chicks are solely provided terrestrial foods, they eat it in amounts such that they achieve normal development.


Figure 2: The data points are almost impossible to see due to the boxplot fill. I would suggest only colouring the outline of the boxplot and colouring the data points as per the treatment group they belong to.


Figure 3: It would be informative to make this a 2-plot figure with the second plot showing the chicks’ second food preference as this is relevant to your last set of analyses.


Figure 4: This figure would be even more informative if each no and yes bar was coloured by the proportion of 1st and 2nd choice being fish or cat food (for the high protein food) and mussel or bread.
For example, perhaps the first “no” bar with a frequency of 5 has a 2/5 proportion of bread and 3/5 proportion of mussels. Y=1-2 would then be coloured to represent bread and y=2-5 would be coloured to represent mussels. This would help explain your last set of results where you do not get a preference for high-protein food because chicks likely preferred marine food as their 1st and 2nd choices.

·

Basic reporting

Using an experimental setup with chicks raised in aviaries Inzani et al. provides evidence that soon after hatching Herring Gull chicks develop a preference for fish over cat food and cooked mussels and do not favour bread. Globally the manuscript is well written and figures and tables are nicely presented. However, statistics can be improved and some flaws in the text and figures should be addressed. I think it potentially makes a sound contribution to science, but I have some concerns about the experimental design as well as some conclusions linked to this that should be addressed.

I find the abstract very focussed and to the point. Instead other parts, and especially the discussion are rather lengthy and not always relevant for the message of the paper. Also I have some problems with the artificial contrast between anthropogenic and natural food items in which anthropogenic food is considered as junk-food per se. I think that definitions in this respect should be carefully considered and used consequently (see specific comments below). In the introduction (but also in other sections) you emphasize possible fitness consequences of new feeding habits in wildlife, more specifically urban foraging. To put it very generalised you try to convince the reader that animals that animals replace their natural high-quality food items with poor-quality anthrophonic food, the junk-food hypothesis. You suggest that many gull species, including Herring Gulls used to feed on marine resources and but nowadays switched to terrestrial resources that are not natural but anthropogenic. You make a distinction between natural marine food and anthropogenic terrestrial food (or sometimes terrestrial anthropogenic food, urban food). However, the difference between natural, high quality, marine food and terrestrial, low quality, anthropogenic food is not always clear in Herring Gulls and I would suggest to chance the terminology in “marine versus terrestrial”. In fact in Europe increased fishery activity (anthropogenic!) is held responsible for the strong increase in many gull populations including Herring Gull during the second half of the twentieth century as discarding made energy rich prey (not directly junk-food) more available to foraging gulls. In other words: natural food of marine origin which was not so much part of the original diet of Herring Gulls, was made available by anthropogenic practices. This hopefully explains my point, that there is a difference between natural, original and non-anthropogenic. Your marine group, for example, was fed with cooked mussels which in my opinion is a very anthropogenic food source since it is processed by humans and probably additives are added. Nevertheless wild mussels (natural) do belong to the original food choice of Herring Gulls. Similarly the cat food you used in your experiments contains natural ingredients (meat), but is strongly processed and non-original for gulls. From an evolutionary perspective one might expect that young naïve gulls choose for original marine food sources such as intertidal organisms, crustaceans and fish, which in fact is what you find. From an nutritive perspective, one might expect that the chicks would choose for energy-rich, protein-rich food source. For the above reasons I think that introduction should be a little bit reworked, checked for unfounded statements (natural versus anthropogenic) and that you simply use the terms “terrestrial” and “marine” instead.

Some examples:
Line 131 “anthropogenic or terrestrial foods” in contrast to “natural intertidal or marine prey”. Are terrestrial earthworms, insects and mammals like moles, which Herring Gulls frequently feed on not natural?
Line 162-165: I would change “Our dietary manipulation reflects the extremes in the range of foods that gulls have been observed to provision their chicks in the wild (Pierotti & Annett, 1991; Sotillo et al., 2019), to explore whether chicks provisioned with anthropogenic terrestrial foods (simulating urban-foraging parents) would develop a preference for those foods too” to “ Our dietary manipulation reflects the extremes in the range of foods that gulls have been observed to provision their chicks in the wild (Pierotti & Annett, 1991; Sotillo et al., 2019), to explore whether chicks would develop a preference for those foods”

Also you should avoid using the terminology “junk-food” as a substitute for all terrestrial food. In your case you use cat food as one of the terrestrial food sources. Indeed cat food is sometimes use by herring gulls to feed to the chicks, but I think you will offend a lot of cat lovers when you call this junk-food. In fact you do show yourself that it is rich of energy and proteins.

Experimental design

My main concerns have to do with the experimental setup and the communication about the results. Firstly, I have some doubts about the difference between the two dietary groups, which should be dealt with. This might be a matter of more precise wording, but if not it could have consequence for the plausibility of the results as well as the title. Secondly, I do think that one of the major findings of the paper is largely ignored. Inzani et al. actually demonstrate that the gull chicks in their experiment developed a specialisation for fish within 5 days from entering the facilities. In fact when first tested at arrival at the facilities the chicks showed no specific food preference. In their manuscript this first test included in the Methods and regarded as a baseline test, however I think it is a crucial finding that must be addressed thoroughly .

Line 256-262: this part of the experimental setup is somewhat confusing and I think it somewhat undermines one of your main research questions and conclusions. I keep wondering and I think it is crucial to know, whether food consumption of the experimental chicks actually resembled the diet composition that was offered to them. Since the food was given ad libitum and always some part of diet consisted of marine food, how can you be sure about the actually food intake of individual chicks? In the manuscript you assume that food intake of the entire group was at the ratio 80:20, but could it be more equal for individual chicks because they selected certain types of food? Could it, for example, be possible that some individual chicks (or maybe even all chicks) within the terrestrial group consumed mainly marine food items and largely ignored the presented terrestrial food items? In other words: did all chicks consume the presented food in a 80:20 ratio and can you rule out that there was variation in the degree of individual specialisation within an experimental group? Even within the marine group it can be possible that some chicks consumed more terrestrial food than others or, less likely, even specialised on terrestrial food without you knowing it. It is clear that there was a difference in the food offered to the chicks, but was there indeed a difference in consumption between the terrestrial and marine group? Your statement in line 262-264 that terrestrial food was often neglected, is not very reassuring in this respect. Did you maybe take blood or feather samples for SIA or video footage to show differences in individual intake ratio’s or how can you otherwise be sure of a true difference between the groups? If you are not sure about the actual intake of individual chicks I am afraid you must weaken your conclusions about the absence of dietary conservation.

Validity of the findings

Discussion and line 54-56 of the abstract – Do your findings indeed suggest that “the consumption of bread and other anthropogenic foods seen in wild populations of mature gulls is unlikely to be a result of early-life exposure to those foods.”? In this respect I am very intrigued about the fact that the chicks soon after intake showed no preference for a certain food type (Line 318-321 of the Methods). Obviously they developed a preference for marine food (especially fish) within 5 days after the first test. This baseline test is largely ignored in the rest your paper as you only focus on the tests performed between day 5 and 35, but is very crucial. Could it be that the chicks developed a preference for marine food because eventually all chicks had access to marine fish (maybe some more than others, but in theory all chicks could potentially taste and consume fish)? Would the result be different if would have a group that was totally deprived from marine fish from the beginning onwards? In wild chicks there are large differences in diet composition depending on the dietary specialisations of its parents. This enormous individual variation in foraging specialisation and diet is very typical for Larid gulls and as you mention yourself it is very relevant to know whether this results from differences in early-life experiences (either as a chick or postfledging) or from an inherent genetical component. In your manuscript you try to convince the reader that early life-experience is not important but that the preference for high protein fish is rather an adaptive mechanism to cope with high energetic and nutritive requirements during early growth. But in contrast your experiment suggests that a food preference might develop very fast in the first days post-hatching. If that is the case food preference of wild individuals will strongly depend on the diet during first few days post-hatching and thus on the dietary specialisation of the parents. This should be dealt with in your manuscript.

Additional comments

Methods
Line 171-173: this is not clear. Would you expect the terrestrial raised chicks to progressively prefer terrestrial high-protein food or would you expect chicks to chose for high-protein food irrespective of the group (marine/terrestrial) in which they were raised?
Line 194: So some chicks entered the study in a later stage which raises questions: what did these chicks eat during their stay at the vet and did you test whether this influenced your study outcome?
Line 197: Were these chicks temporarily separated from the group and what diet did they receive in the meantime? Did this influence your results?
Line 198: Looking at the graph in the supplements I think this must read “Chicks were weighed daily until ca. 30 days of age, then at least once weekly …”
Line 199: at least weekly?
Line 201-203: move to “Ethics”
Line 218-219: better use "outdoor aviaries" and "outdoor flight pens" so that it is easier for the reader to understand what you mean by "outdoor enclosures" in line 220.
Line 225-226: This is an overkill of references only to show how tarsus length was measured. Only referring to Caravaggi et al. is sufficient.
Line 229-231: Handling time just before release is only relevant for ethical reasons (move to ethics?) not for your study design or results. Instead it would be much more welcome if the reader would be informed about handling time during daily manipulations of your chicks.
Line 255: do you mean “or” instead of “and”? Or did you always present both food types at the same time?
Line 256-262: I think this part of the methods needs some more detail and clarification since it is a crucial part of your experimental setup. To me it is very confusing and raises questions. If I understand it right this is what happened: You presented your chicks 5 ad libitum meals per day using different ratio’s of marine and terrestrial food. The ratio you mention 80:20, is indeed possible by changing the food four times after you first presented food in the early morning (for example 4 times terrestrial food and 1 time marine food to reach a ratio). However, how did you reach a 80:20 ratio in older chicks when you changed food only 3 times a day (line 259)? Did you each time offer high and low protein food (either terrestrial or marine). If yes did you offer it in different trays or in a mix? Or maybe you even used a totally different method in which case your description should be clarified? Also important to mention whether or not you removed non-consumed food items before presenting new food to the chicks. Please be more precise here because it is very important to understand the impact of your results (see my comment ….).

Statistics
Lne 364-369. Here sample sizes are too small for chi-squared tests, so either you should use a Fisher's exact test or perform a GLMM with binomial link on all data with “test trial day” as a random variable to avoid pseudoreplication.


Results
Line 374-375: “Chicks’ overall ….. (Table 1)”. Can be deleted.
Line 375-377: “As shown in Table 1 and Figure 1, chicks preferred to consume fish first and appeared to avoid bread and cat food, while for mussels they showed no significant preference or avoidance.” I have two points here that I think are equally important. In the first place the Table 1 does not show the same outcome as Figure 2. According to Table 1 chicks chose 2 times for cat food in the Day 10 trial, while Figure 1 suggest that this was 4 times (2 times in the marine group and 2 times in the terrestrial group). Which is right and does this have implications for your test results? In the second place I am not convinced about your statement about preferences for certain food types. You only tested for an overall significance between the food types, but you did not perform a post hoc test to find out which cells from the contingency table are actually different from their expected values. Furthermore I think that sample sizes are too small for a chi-squared test and you should consider a Fisher's exact test or a GLMM with binomial link in which you use “test trial day” as a random effect.
Line 381-382: “Chicks’ first food choices reflected their food preferences in terms of the amount of each food type that they consumed (Figure 2).” This is not what is shown in Figure 2. I can see your point when comparing Figure 1 and Figure 2, but not when only looking at Figure 2. However, when doing so it seems to me that there is not always a one to one relationship between first food choice and amount of food consumed. It might be true for fish and bread, but not so much for cat food and mussels. For example, the rather high intake of cat food in the day5 and day10 trials (Figure 2), is not expected given the avoidance of this food type as a first food type. Also, the rather high mussel-intake of chicks at day 35, does not reflect the ignorance of mussels in the first trial of the terrestrial group. Can you maybe test your statement?
Line 384-385: “Cat food consumption decreased with increasing time spent in captivity”. This is the only indication of a learning effect which seems very important within the context of your paper. I would like to see a statistical test here, to test for the time in captivity and treatment group on the amount of food consumed (using Chick ID as a random factor).
Line 388-389: “Chick ID explained little variation when included as a random effect in the GLMM model described”. Please show this information in the supplementary Table S2. I assume that Chick ID is excluded as a random effect in the test results shown in Table S2?
Line 394-395: Why do you suddenly include the chicks’ second food choice? There is no mentioning of this in the pervious sections.

Discussion
In general I think the discussion is too long and not very focussed.
Line 408: I think that you often did not test for individual differences (rather for supposed dietary of the group) as for most of your statistical tests Chick ID was included as a random factor.
Line 409-411: “We found that herring gull chicks unequivocally preferred marine food, specifically fish. Preference for cat food and mussels was not clear nor consistent.” The second sentence suggests to me that mussels are not included in “marine food” of the previous line. The message however is that in each age group more than 80% of the first choices were directed to marine food (fish or mussels). Maybe a slightly different word choice?
Line 411-412: “This finding supports previous studies showing that wild parent gulls that provide a more marine-based diet … ”. I believe that a preference of captive chicks for marine food might explain why wild parents preferably provide fish to their chicks, but it certainly does not explain why the chicks grow better on a fish-enriched diet. Please use other wording here.
Line 414: Change “The more natural marine prey….” in “Marine prey….”
Line 421-422. Delete “This suggests that a preference for fish is adaptive for chick growth and development.”
Line 424-444: Here you suggest that animals may be forced to feed on anthropogenic food sources during the breeding season due to limited foraging ranges, and switch to more natural resources in winter. However, in most gull species including Herring Gull some (and often most) individuals chose to feed on anthropogenic food sources (rubbish dumps, towns, agricultural areas, discards etc.) in winter. Very often they even switch to a more marine oriented diet during chick rearing. Very much the opposite of what you suggest.
Line 449-451: To what extent could the preference for high protein food be explained by a strong preference for fish (which in turn maybe for other reasons than protein)?
Line 453-457: Peak energy demands of growing Herring Gull chicks are much later than age 15! This can thus not explain your results.
Line 461-483: I find this paragraph not very relevant for your study. Here the “anthropogenic is bad and natural is good” intention is very obvious, while the truth is much more complicated. Some food that is facilitated by humans might be of equal (or sometimes even better) quality as natural food. Increased availability of fishery discards in the second half of the twentieth century is in fact held responsible for the exponential growth of many gull populations and is deemed profitable for chicks and adults. But also some terrestrial food items are probably not very bad for the development of the chicks. What is important is that you initially find no preference for a certain food type (marine vs. terrestrial) in naïve chicks that entered the facilities. And obviously a very steep learning curve towards a preference of marine food (within 5 days chicks did preferred marine food - always > 80% in each trial).

References
Please check the references to surnames starting with “van”. Van Donk, Van der Meer and Van Den Bosch are sometimes found under the “V” and sometimes not.

Reviewer 3 ·

Basic reporting

The paper is well-written and largely clear. The methods section could be more concise and unambiguous - esp. with the inclusion of a figure demonstrating the experimental process. The literature is well reviewed and the figures are adequate.

The results are relevant to the hypothesis but they could be more clearly set out and built on in the discussion - specifcally:

Line 174. Wy might you expect individual differences in food preferences, and why these would be consistent overtime? This might be clearer if you restructure your hypothesis to better include your reasonings behind them.

Experimental design

1. The research question is well stated, however, my main comment is that there is a lot of detail in describing the experimental preparation and set-up which is required. As it is currently presented it is very difficult to determine exactly what was involved and whether this could be replicated. It would be beneficial to the reader to ensure the ‘Experimental protocols’ section is as clear and concise as possible and to add a diagram/flow chart to better visualise each step. In particular it was clear how chicks were assigned to the Marine or Terrestrial treatment group (lines 321). Was this random – and if so how? It is interesting that in the initial baseline food preference tests the chicks showed no preference for either first or cat food but in the following experimental tests chicks appear to choose fish over cat food – do you have any thoughts on why this might be?

2. Another aspect which I had questions / caused confusion was the different ages of the chicks at each experimental stage. You state on line 190 that the chicks were aged 1 to 26 days when they were admitted to captivity. Therefore, if the experiments were conducted on days 5, 10 15 etc after the gulls had been admitted then the age range of chicks within each experimental test is large. Did you test whether chick age (rather than how long they had been in captivity – which I know will be related to chick age) had any influence on their choices in your analysis? Please can you also provide details in the supplementary material on the age of each chick when it has admitted i.e. to better understand if most chicks were less than 5 days or older? One thing that you didn’t acknowledge was whether what the chicks had been fed before they were admitted might have influenced their choices. This should be briefly discussed as this is an unknown that cannot be accounted for. Finally, do you have information on where these chicks came from i.e. I assume they come from more urban nesting environments where people come into contact with ‘orphans’ rather than from traditional coastal colonies?

3. Section ‘Chick husbandry during captivity and release’ is really important to include to understand the animal welfare / ethics of how these gulls were kept. However, the text from lines 205 – 245 are not particularly relevant to the experiments so could be added to Supplementary Material. Although it would be good to retain that chicks were individual marked in the main test to determine how you could follow each chick through the process.

4. Figures 3-4 could be added to the supplementary material.

Validity of the findings

The discussion doesn’t clearly link back to your main hypothesis/research questions and whether these were met or not, and why. The discussion is also rather wordy and would benefit of being made more concise to better emphases your key results and how they link to what other studies have found / increased our knowledge.

Thank you for providoing the underlying data. I t was not immediately clear what all the data columns in the observer_comparison file refer too. Also are the first and second food columns for the first observer and frankie.first.choice the second observer? Did the second observer not have data on the second choice food?

For both data spreadsheets it would be valuable to have the column names described in full, i.e. in another tab.

Additional comments

This is an interesting paper looking at the food preferences of herring gull chicks bought into a rescue center. The food preferences of these chicks were experimentally tested to determine whether they showed an innate preference for certain high or low protein food choices from a marine or terrestrial origin – or whether choices were influenced by what they had been predominantly fed (marine or terrestrial) before the experimental trails. This is a very useful additional to the literature in better understand diet choices of a generalist species which inhabits / exploits habitats influenced by anthropogenic change.

My main comments are provided in the above sections.

More general comments are provided below.

Line 93. This is a little repetitive of above on line 77. One option could be to re-order this part of the introduction to talk about carnivores first - then come onto seabirds as that is the focus of this paper.
Line 104. This sentence on Silver Gulls feels out of place in this paragraph. It would be better below when you talk about gulls specifically.
Line 136. The way in which this sentence starts feels clunky and it isn’t entirely clear what you are trying to say so suggest rephrasing.
Line 211. This sentence is repetitive to that on line 207.
Line 244. Here you say the mean age of released chicks was c71 days, but on line 223 you say chicks were deemed fit for release a c 50 days? Is there a reason for this discrepancy?
Line 246. More info. required on the age of chicks in each groups – or all similar? As similar period of breeding season?
Line 312. You include that you recorded whether the chicks were tame or not – was this information used in your analysis in any way? It might be interesting to determine whether this measure of ‘personality’ influence food choices.
405. Although the sample size was only really lower for the 35 treatment?
Table 1. Be consistent with Fig. 2 and include significant results in bold.
Line 528. Although you cannot know what the chicks were being fed before they arrived?
Line 542. Apologies if I missed this, but looking at these two references I couldn’t see any evidence for chicks actively refusing anthropogenic food leading to increased begging?

---

## Round 0.2 · Minor Revisions

Dear Authors,

I extend my heartfelt gratitude for your prompt response to the feedback provided by the three reviewers. It is encouraging to note that they concur on the necessity for minor revisions before publication. Primarily, their suggestions center around enhancements in the introduction and the materials and methods sections. I eagerly anticipate receiving your revisions at your earliest convenience, thereby facilitating the readiness of the manuscript for publication.

Warm regards,

Armando Sunny

·

Basic reporting

I thank the authors for thoroughly reviewing their manuscript based on the comments made during the first round of revision. This current version of the manuscript is adequate for publication. My main concerns regarding referencing issues have been adequately addressed and the manuscript is now much more succinct and focused. Regarding the introduction, I only have two minor comments that the authors can take or leave as they see fit:

The first paragraph of the introduction appears somewhat removed from the premise of this paper as it talks about the difficulties associated with living in an urban area very broadly and does not necessarily relate with food choice/foraging option in cities. I appreciate that the authors are trying to keep the paper within a certain word range, therefore I think this aspect of the paper could be trimmed to focus solely on foraging/nutrition in urban settings.

L167: Did you mean “consistent intra-individual preferences” here instead of “consistent inter-individual differences”? If this sentence refers to testing for consistent individual preferences, I would suggest changing the wording accordingly.

Experimental design

The methodological details that were missing from the first version of the manuscript (RE: where the chicks came from/their possible diet prior to being rescued & the effect of their dietary treatment on their growth curve) are now described. I only have 2 points remaining that I think would improve the manuscript if they were addressed:

L209-211: Since 2 chicks were removed from the study for a certain period of time (unstated) before being returned, how did their diet change during this time? What was their assigned dietary group, did they remain on the same diet while sick, had they already performed preference test trials before being taken out of the study due to their illness – and if so, did their food preference change post-illness? More information about your decision to keep these 2 subjects in the study despite the illness would be required. If those 2 subjects are taken out of your sample size, do your results remain the same? If this information makes the manuscript too wordy, I would suggest only giving a general overview in the main text (i.e. whether including or excluding these birds from your analyses changed the results) and going into further details in the supplementary material.

L388-389: The results would be even more informative if the median growth curves (and IQR) for terrestrial vs marine-raised chicks were included in the graphs.

Validity of the findings

The authors have added the additional analyses required and tempered their conclusions accordingly such that their findings are well-represented and discussed adequately.

Additional comments

I congratulate the authors on a very interesting study that compellingly shows the dietary preference that HERG chicks exhibit towards fish (and they aversion towards bread) despite having access to a variety of foods in different ratios.

·

Basic reporting

no comment

Experimental design

no comment

Validity of the findings

no comment

Additional comments

I am very satisfied with the changes made by Inzani et al. to their manuscript 'Early-life diet does not affect preference for fish in herring gulls (Larus argentatus)' and with the way they have addressed the quite substantial comments from the reviewers. I actually have not much further comments.

What I still miss is a reflection on the fact that wild chicks do indeed accept bread from their parents. The manuscript now suggests that a) Herring Gull parents switch to an almost fully marine diet when the eggs hatch because the chicks refuse bread (lines 512-515 of the pdf) and that b) a preference for bread only develops later in their lives (lines 481-482 of the pdf). That doesn't seem totally accurate to me. There are several publications indicating that HG parent sometimes do feed bread to their chicks and that the chicks do consume bread. I also observe this regularly in our colonies. I quickly checked my own data on regurgitations of HGs during handling. When selecting only birds that actually regurgitated food (many do not), my data shows that 10 out of 42 adults caught in our colonies regurgitated bread, while 2 out of 58 chicks did so. This indeed indicates that adults may feed on bread more often than chicks, but at least some chicks do consume bread. A brief paragraph discussing the difference between the findings of Inzani et al. in captivity and the diet of wild chicks would therefore be appropriate.

Reviewer 3 ·

Basic reporting

No comment

Experimental design

No comment

Validity of the findings

No comment

Additional comments

Thanks very much to the authors for addressing my, and the other reviewers, previous comments. The manuscript is much improved and clearer to follow, with the discussion better reflecting the results. I do have some further comments, mainly to do with the introduction (please see specific comments below).

Line 45. This question is not really tested or answered by your study. To manage expectations I recommend adding a sentence here outlining what your study does try to address i.e. whether chicks do have a preference which might then be a step in indicating whether parental switches is driven by chicks preferences. It should also be noted that the switch to marine food does not occur in all colonies i.e. if that resource is not available.
Line 59. I'm still not convinced that your results indicate that chicks reinforce the provisioning on marine food - I think this can only be tested in the wild to see if chicks do refuse other food types? So you could add that as the next step here to determine if this is the case?
Line 81. Coastal areas rather than sea cliffs per se? I understand what you are mean here i.e. if urban areas encroach onto sea cliff tops - so unlikely that birds will nest there but the cliffs themselves are usually not urban?
Line 90. Change 'are' to 'can be easily accessible'. As measures are put in place at some locations to prevent access to such food from gulls.
Line 120. Needs a link here between urban areas and other areas where anthropogenic activities result in food for species away from urban areas i.e. pelagic seabirds that you talk about here and discards.
Line 224 - 234. A key point here that should be mentioned is that in most of these studies it depends on what marine and terrestrial/anthropogenic foods are available within the foraging range of each colony - so results are very site specific. i.e. are they comparing fish with grain, or intertidal invertebrates with anthropogenic food waste (chips / burgers etc...). Both these examples compare marine v terrestrial/anthropogenic food but the quality of each is very different (i.e. in terms of energy, fat, nutrients etc).
Lines 239 - 246. There are a large number of references here and I understand why given the range of diet of gulls but I recommend reducing the number to several key ones.
Lines 253 - 25. Is this true though - I would argue it is also / more influences by the availability of resources within the foraging range of the colony (at least for breeding adults). So instead change the wording along the lines of 'In part' rather than 'likely to be particularly'. I also couldn't find anything obvious in the provided references to suggest that gull's foraging preferences / strategies were influenced by their parents / conspecifics?
Lines 253 - 273. I struggled with this paragraph as the links between each point are difficult to follow. Please consider restructuring / rephrasing to better emphasise what you want the reader to take away here.
Line 265. Though traditionally Herring Gulls forage in intertidal areas more than at-sea catching fish – therefore I recommend rephrasing this sentence.
Line 272. There is a bit of a jump here - do we know how natural nesting chicks acquire their dietary preferences?
Line 278. Move 'a predominantly' to before (i)
Line 299. Include here that these chicks were GPS tagged as part of another study - i.e. to manage expectations as the GPS data was not used in this study.
Line 308. Include 'rehabilitation' in full?
Line 730. Remove 'for' at end of the sentence
Line 744. There are a lot of reference to fish being the most preferred / consumed diet by Herring Gulls which may be the case in some colonies. However, traditionally intertidal prey is / will be more important. i.e Many diet and tracking studies show that Herring Gulls spend little time at sea scavenging or catching fish. Therefore, where individuals do rely on anthropogenic food like bread it might be because there are also few foraging opportunities in intertidal habitats.
Line 753. But this has been looked at by some studies in the short-term for some species where anthropogenic diets result in poor body condition.
Line 777. As mentioned by a previous reviewer, do chicks in the wild refuse food? I have not observed this, or come across any instances of this. Instead chicks eat what ever is regurgitated - so I am not convinced by this argument. You could add that this should be looked at in the wild to determine if this does occur?
Line 857. It should be included here that this isn't a unbiased comparison of terrestrial versus marine food as the terrestrial food had supplements added to it.

---

## Round 0.3 · accepted · Accept

Dear Authors,

I am pleased to inform you that your manuscript has been accepted for publication in PeerJ. Thank you for choosing our journal to share your fascinating research.

Best regards,

Armando Sunny

·

Basic reporting

Congratulations with your excellent revision of the MS. I have no further comments.

Experimental design

No comment

Validity of the findings

No comment

Additional comments

no comment

Reviewer 3 ·

Basic reporting

na

Experimental design

na

Validity of the findings

na

Additional comments

Thanks very much to the authors, I have no further comments!